# Learning Frequency-Adapted Vision Foundation Model for Domain Generalized Semantic Segmentation

**Qi Bi**[1,*] **Jingjun Yi**[2]**, Hao Zheng**[2✉]**, Haolan Zhan**[3]**, Yawen Huang**[2]**,**
**Wei Ji**[4]**, Yuexiang Li**[5✉]**, Yefeng Zheng**[1✉]

[1]Westlake University, China, [2]Jarvis Research Center, Tencent Youtu Lab, China,
[3]Monash University, Australia, [4]Yale University, United States,
[5]University of Macau, Macau
`howzheng@tencent.com`, `yuexiang.li@ieee.org`
`zhengyefeng@westlake.edu.cn`

## Abstract

The emerging vision foundation model (VFM) has inherited the ability to generalize to unseen images. Nevertheless, the key challenge of domain-generalized semantic segmentation (DGSS) lies in the domain gap attributed to the cross-domain styles, *e.g.,* the variance of urban landscape and environment dependencies. Hence, maintaining the style-invariant property with varying domain styles becomes the key bottleneck in harnessing VFM for DGSS. The frequency space after Haar wavelet transform provides a feasible way to decouple the style information from the domain-invariant content, since the content and style information is retained in the low- and high-frequency components of the space, respectively. To this end, we propose a novel Frequency-Adapted (FADA) learning scheme to advance the frontier. Its overall idea is to separately tackle the content and style information by frequency tokens throughout the learning process. Particularly, the proposed FADA consists of two branches, *i.e.,* low- and high-frequency branches. The former is able to stabilize the scene content, while the latter learns the scene styles and eliminates its impact to DGSS. Experiments conducted on various DGSS settings show the state-of-the-art performance of our FADA and its versatility to a variety of VFMs. Source code is available at `https://github.com/BiQiWHU/FADA`.

## 1  Introduction

Most existing semantic segmentation tasks assume that the training and inference images follow the independent and identical distribution (i.i.d.) [12, 13, 36, 35, 34, 81, 54, 44, 74], which is far from reality. Domain-generalized semantic segmentation (DGSS) aims to infer robust pixel-wise semantic predictions on arbitrary unseen target domains when a segmentation model is trained on the source domain (as illustrated in Fig. 1a). Compared with general domain generalization tasks, the feature distribution discrepancy between the source domain and unseen target domains in the context of DGSS holds some unique factors. Specifically, the cross-domain images in DGSS usually share the same content information (*i.e.*, common semantic categories in driving scenes), while the cross-domain styles (*i.e.*, urban landscape, weather, lighting conditions, *etc*.) mainly account for the feature distribution difference [65, 49, 18, 11, 72, 6, 23, 43, 24].

Existing DGSS methods can be summarized into three categories. The first category intends to decouple the style information from the scene representation [55, 30, 14, 56, 78, 71, 58], but does not

---

[*]Qi Bi is affiliated with University of Amsterdam. This research was conducted with Westlake University and Tencent Youtu Lab.

38th Conference on Neural Information Processing Systems (NeurIPS 2024).

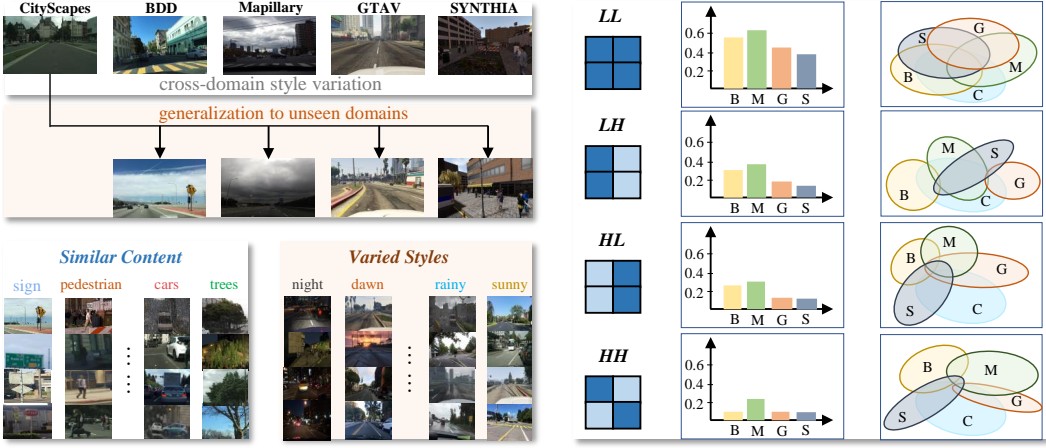

(a) Domain Generalized Semantic Segmentation      (b) Domain Distribution after Haar Transformation

Figure 1: (a) The key challenge of domain generalized semantic segmentation (DGSS) lies in the stability of the scene content, while the domain gap is caused by the style variation. (b) Analysis of frozen VFM features after Haar wavelet transform. We compute the correlation coefficient between the CityScapes source domain (C) and BDD, Mapillary, SYHTHIA, GTA5 unseen target domains (B, M, G, S). The low-frequency component exhibits a higher correlation and smaller domain gap. In contrast, the high-frequency components exhibit a lower correlation and a larger domain gap.

ensure a strong content representation ability. The second category, on the contrary, directly focuses on the content representation ability regardless of the cross-domain style variations [19, 4, 7]. Such methods were recently developed much owing to the stronger pixel-wise representation ability of mask attention [12, 13]. The third category focuses on enriching the styles as much as possible during training [82, 41, 83]. However, the content representation ability is rarely taken into consideration.

The emerging vision foundation model (VFM) [62, 39, 20, 37], with strong generalization ability inherited from a large quantity pre-trained images, provides a new possible paradigm for DGSS. Under the realm of parameter-efficient fine-tuning, the hypothesis, *i.e.*, a foundation model relies on the low intrinsic dimension [42, 1] to adapt to the downstream tasks, exemplified by low-rank adaptation (LoRA) [28], has shown great success, where the key idea is to inject trainable rank decomposition matrices into each layer. More recently, the low-rank adaptation paradigm has demonstrated to be feasible to fine-tune the VFM for DGSS [69]. Unfortunately, the style-invariant properties of VFM, which is a fundamental problem for the extraction of domain generalized semantics, remain unexplored.

An ideal style invariant low-rank adaptation is supposed to discern the subtle low intrinsic dimension while does not pose shift on the fragile and frozen VFM features. The prior DGSS methods learned style invariance from the fully trained image features (e.g., instance normalization [55, 30] and instance whitening [14, 56, 71, 58]), which are more likely to collapse and less feasible. Therefore, we adapt the low-rank adaptation to the frequency space, where the style and content have been separated to high-frequency and low-frequency components [29, 46, 40, 77, 67], respectively.

In this paper, we present a novel **F**requency-**Ada**pted learning scheme, dubbed FADA, to push this frontier. Its conceptual idea is to adapt the style and content representation from the VFM features separately in the frequency space. After transforming the frozen VFM features to the frequency space by the Haar wavelet transform [60], the low-frequency branch exploits the scene content from frozen VFM features by the learnable low-frequency tokens. In contrast, in the high-frequency branch, the high-frequency token features are implemented using the instance normalization operation, so that the representation becomes invariant to the scene style.

Notably, the proposed FADA introduces two new research lines. Firstly, the possibility of learning low-rank adaptation in the frequency space is explored, which can further benefit other VFM downstream tasks strongly related to the style and content representation. Secondly, it demonstrates the potential of harnessing the Haar wavelet transform for DGSS, which can also inspire advancements in general visual domain generalization.

Concretely, our contributions can be summarized as follows:

- We propose a **F**requency-**Ada**pted learning scheme, dubbed FADA, to fine-tune VFMs for domain-generalized semantic segmentation.

- The proposed FADA, aided by the Haar wavelet guidance to mine the style-invariant property of VFM, is versatile to a variety of VFMs.

- Experimentally, the proposed FADA significantly outperforms the state-of-the-art DGSS methods, and yields an improvement up to 2.9% mIoU over the contemporary REIN [69].

## 2   Related Work

**Domain Generalization** handles the challenging setting where the feature distribution of an arbitrary target domain is not identical to that of the source domain. It has been extensively studied in the past few years. Multiple techniques, to namely a few, optimal transport [22, 79], batch normalization [66], causal inference [48, 47, 25], discrepancy regularization [80, 68, 17, 5], and uncertainty modeling [59, 70], have been proposed. Furthermore, domain generalization via unsupervised learning [26, 27] or from a single source domain [61, 59, 84, 75] has also been recently studied.

**Domain Generalization by Frequency Decoupling** has drawn increasing attention. Its general idea rests in that the style and content have been demonstrated on high-frequency and low-frequency components [29, 46, 40, 77, 67, 7, 6], respectively. Most of these methods implement Fast Fourier Transform (FFT) to transfer the image to the frequency space, and then represent the style and content by the amplitude (high-frequency) and phase (low-frequency) components, respectively. However, *to the best of our knowledge*, 1) leveraging Haar wavelet for domain generalization; and 2) enhancing the generalization ability of VFM features via frequency space have been rarely explored. Compared with other frequency analysis methods such as FFT, the orthogonal property of Haar wavelet basis leads to a stronger decorrelation [50]. In the context of domain generalization, it indicates a better separation between low- and high-frequency components and deserves exploration.

**Domain Generalized Semantic Segmentation** (DGSS) in the CNN era either decouple the style information [55, 30, 56, 14, 58, 71, 78] or enrich the style diversity [41, 82, 83, 45, 52]. With the rapid development of Vision Transformer (ViT), recent DGSS methods usually leverage the masked attention mechanism [12, 13] to enhance the content representation [19, 8, 7]. Later, the masked attention is used to decode the frozen contrastive image-text pre-trained features [33], and REIN [69] fine-tunes the VFM under the low-rank adaptation paradigm. However, the style invariant properties of VFM, which are the key of the DGSS representation, remain unexplored.

## 3   Preliminary

### 3.1   Low-Rank Adapted VFM

To fine-tune a VFM with parameter efficiency, the low intrinsic dimension [42, 1] assumes that a VFM relies on the intrinsic low-dimension in the frozen VFM features to adapt to the downstream tasks. Inspired by this, the low-rank adaptation (LoRA) paradigm [28] is devised to inject trainable rank decomposition matrices into each layer. Given a VFM with $N$ sequential layers (denoted as $L_1, L_2, \cdots, L_N$), each layer corresponds to a pre-trained weight matrix of $W_1, W_2, \cdots, W_N$, *i.e.*, $W_i \in \mathbb{R}^{c \times c}$. The frozen features from layer $L_i$ are denoted as $f_i \in \mathbb{R}^{c \times n}$. Particularly, for the first layer $L_1$, $f_1$ is generated by $f_1 = W_1 x$. Here $x$ denotes the image embedding, $n$ denotes the patch number, and $c$ denotes the channel size.

Denoting the learnable weight matrix as $\Delta W_i \in \mathbb{R}^{c \times c}$ and the input of layer $L_i$ as $f_i$, the feature propagation from the layer $L_i$ to $L_{i+1}$ can be formulated as $f_{i+1} = W_i f_i + \Delta W_i f_i$. Then, we assume that the learnable weight matrix $\Delta W_i$ can be formulated as a low-rank decomposition, *i.e.*, $\Delta W_i = BA$, where $B \in \mathbb{R}^{c \times r}$ and $B \in \mathbb{R}^{r \times c}$ ($r \ll c$). More recently, REIN [69] specifies this paradigm in DGSS by transferring the learnable matrix term $\Delta W_i$ into a learnable token $T_i$ followed by a MLP $\mathrm{M}_i(\cdot)$, denoted as $f_{i+1} = W_i f_i + \mathrm{M}_i(T_i(W_i f_i))$, where $T_i \in \mathbb{R}^{m \times c}$ and $m$ is the token length. This modification allows a significance reduction of the token length (from a thousand magnitude $c$ to a hundred or even ten magnitude $m$), which can alleviate *Curse of Dimensionality* and allow each token to be better connected to the instances in an image [69]. Our low-rank adaptation is implemented on $T_i$, *i.e.*, $T_i = A_i B_i$, where $A_i \in \mathbb{R}^{m \times r}$ and $B_i \in \mathbb{R}^{r \times c}$ ($r \ll \min(m, c)$).

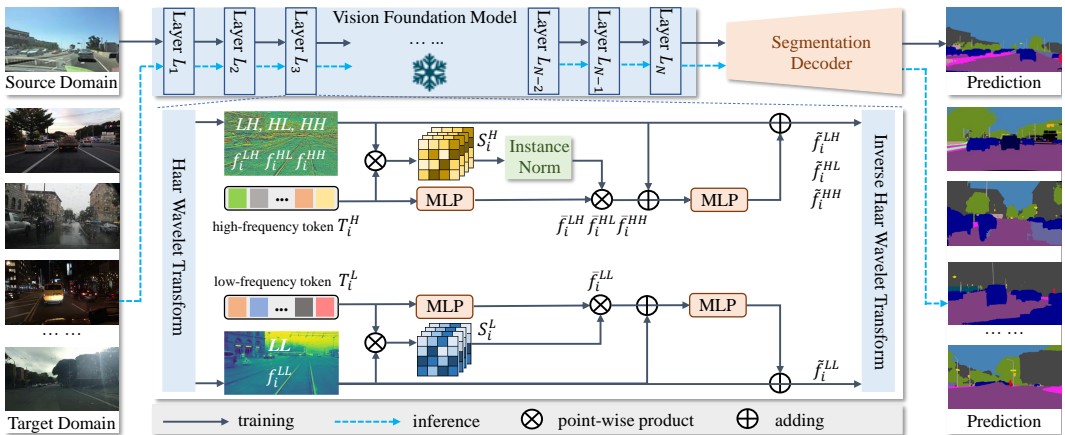

Figure 2: Overview of the proposed Frequency-adapted Vision Foundation Model (FADA) learning scheme. It innovatively incorporates the low-rank adaptation of VFM models on the frequency space, where the low-/high-frequency component contains more content/style, respectively. It consists of three key steps, namely, low-/high-frequency decomposition (in Sec. 4.1), low-frequency adaptation (in Sec. 4.2) and high-frequency adaptation (in Sec. 4.3).

## 3.2 Haar Wavelet Transform

The Haar transform [60] cross-multiplies a function with various shifts and stretches, which has been demonstrated to be effective in applications, such as signal and image processing.

**Definition 1. Haar Scaling Function.** *Given an input signal $x$, the Haar scaling function is mathematically defined as*

$$\phi(t) = \begin{cases} 1 & 0 \le t < 1 \\ 0 & \text{otherwise} \end{cases}. \tag{1}$$

Let $V_0$ denote the space of all functions of the form $\sum_{k \in \mathbb{Z}} a_k \phi(x - k)$, where $k \in \mathbb{Z}$ is an arbitrary integer, and $a_k \in \mathbb{R}$. As each element of $V_0$ is zero outside a bounded set, such a function $a_k \phi(x - k)$ has *finite or compact support*.

**Definition 2. Basis of the Step Function Space.** *Given an arbitrary nonnegative integer $j \in \mathbb{Z}_0^+$, Let $V_j$ denote the step function space at the level $j$, which is spanned by the set*

$$\{\cdots, \phi(2^j x + 1), \phi(2^j x), \phi(2^j x - 1), \cdots\}. \tag{2}$$

**Definition 3. Haar Wavelet Function.** *The Haar wavelet is the function $\psi(x) = \phi(2x) - \phi(2x - 1)$.*

For more details of the properties of the Haar transform, please refer to the supplementary material.

## 4 Methodology

Fig. 2 gives an overview of the proposed FADA. After each frozen VFM layer, it consists of three key steps, namely, low-/high-frequency decomposition (in Sec. 4.1), low-frequency adaptation (in Sec. 4.2), and high-frequency adaptation (in Sec. 4.3). Finally, the frequency components are fused together and transferred from the frequency space back to the spatial space, and then fed to the next VFM layer.

### 4.1 Low-/High-Frequency Decomposition

Orthogonal property (in Sec. 3.2) leads to the result of a strong decorrelation [50]. This property after the Haar wavelet function allows a better separation between low- and high-frequency components of input signal than other frequency analysis methods such as Fourier transform. In the context of DGSS, the domain gap is caused by the cross-domain style variation, while the cross-domain content is stable [14, 78, 71, 58]. As it has been well documented that the style and content are predominate

in the high-frequency and low-frequency components, respectively [29, 46, 40, 77, 67], a feasible way to explore the style-invariant properties is to mitigate the variation of high-frequency components in the VFM features. Therefore, the first step is to decompose the frozen VFM feature $\check{f}_i$ into the low- and high-frequency components, respectively.

In our work, we exploit the Haar wavelet transform to decouple the low- and high-frequency components, where four kernels, namely, $LL^{\mathrm{T}}, LH^{\mathrm{T}}, HL^{\mathrm{T}}, HH^{\mathrm{T}}$, are given by

$$L^{\mathrm{T}} = \frac{1}{\sqrt{2}}[1 \quad 1], H^{\mathrm{T}} = \frac{1}{\sqrt{2}}[-1 \quad 1]. \tag{3}$$

As discussed in prior works [60, 3], the $LL^{\mathrm{T}}$ kernel captures the average of the pixel responses, which is more robust to the scene content and therefore preserves more low-frequency components. In contrast, by taking the differences between adjacent pixels into account, the $LH^{\mathrm{T}}, HL^{\mathrm{T}}$ and $HH^{\mathrm{T}}$ kernels tend to preserve the details from the horizontal, vertical and diagonal directions, respectively. These details are more related to the structures, edges, *etc*, which attribute to the style.

For a layer $L_i$, we implement the Haar wavelet transform on the frozen VFM feature $W_i f_i$ by the above four kernels $LL^{\mathrm{T}}, LH^{\mathrm{T}}, HL^{\mathrm{T}}$ and $HH^{\mathrm{T}}$, respectively. The $f_i^{LL}$ component filtered by $LL^{\mathrm{T}}$ captures more scene content, and in contrast the $f_i^{LH}, f_i^{HL}$ and $f_i^{HH}$ components filtered by $LH^{\mathrm{T}}$, $HL^{\mathrm{T}}$ and $HH^{\mathrm{T}}$ capture more style information. This decomposition is computed as

$$f_i^{LL} = (W_i f_i) \otimes LL^{\mathrm{T}}, f_i^{LH} = (W_i f_i) \otimes LH^{\mathrm{T}}, f_i^{HL} = (W_i f_i) \otimes HL^{\mathrm{T}}, f_i^{HH} = (W_i f_i) \otimes HH^{\mathrm{T}}. \tag{4}$$

## 4.2 Low-Frequency Adaptation

In the context of DGSS, a stable scene content representation despite the style variance is important to predict the scene semantics. For each layer $L_i$, the scene content from the frozen VFM features rests more in the low-frequency component $f_i^{LL}$ (in Eq. 4), which is learned in the low-frequency adaptation branch.

Assume we have a low-frequency token $T_i^L \in \mathbb{R}^{m \times c}$, where $m$ is the sequence length of $T_i^L$, and $c$ is the dimension of the frozen VFM feature defined in Sec. 3.1. The low-frequency token $T_i^L$ is used to exploit the scene content from the low-frequency component $f_i^{LL}$, while at the same time following the low-rank adaptation paradigm (in Sec. 3.1).

First, we compute a similarity map $S_i \in \mathbb{R}^{n \times m}$ between the token $T_i^L$ and low-frequency component $f_i^{LL}$ from the frozen VFM feature, which measures the correlation between each element in $T_i^L$ and each patch embedding represented in $f_i^{LL}$, given by

$$S_i^L = \mathrm{Softmax}(\frac{f_i^{LL} \times T_i^{L^{\mathrm{T}}}}{\sqrt{c}}), \tag{5}$$

where $\mathrm{Softmax}$ denotes the softmax activation function.

Then, we project the token feature $T_i^L$ into the feature space of $f_i^{LL}$ by a multilayer perceptron (MLP) parameterized by weight parameters $W_i^1$ and bias parameters $b_i^1$, followed by the point-wise product with the similarity map $S_i^L$. The product with $S_i^L$ allows the token features $T_i$ to better align to $f_i^{LL}$, where the scene content is highlighted. Assume the output is denoted as $\overline{f}_i^{LL}$. Briefly, this process can be mathematically expressed as

$$\overline{f}_i^{LL} = S_i^L \times [T_i^L \times W_i^1 + b_i^1]. \tag{6}$$

Then, we fuse the projected token features $\overline{f}_i^{LL}$ with the low-frequency features $f_i^{LL}$ by another MLP parameterized by weight parameters $W_i^2$ and bias parameters $b_i^2$, followed by the skip connection. Assuming the output is $\widetilde{f}_i^{LL}$, this process can be mathematically computed as

$$\widetilde{f}_i^{LL} = f_i^{LL} + (\overline{f}_i^{LL} + f_i^{LL}) \times W_i^2 + b_i^2. \tag{7}$$

## 4.3 High-Frequency Adaptation

For DGSS, the robustness to the cross-domain style variance is particularly important. Such style difference is usually reflected on the high-frequency components. The Haar wavelet transform

enables the separation of these high-frequency components $f_i^{LH}$, $f_i^{HL}$ and $f_i^{HH}$ (in Eq. 4). Directly eliminating all the high-frequency components seems to be a simple and straight-forward solution, but it also leads to the loss of other information such as structure and object boundary. It may degrade a scene representation and decline the segmentation performance. Therefore, the objective of the high-frequency adaptation branch is to mitigate the impact of cross-domain style variation, not directly removing all the high-frequency components.

As the decoupling of styles does not differentiate whether the high-frequency components are from the horizontal, vertical or diagonal directions, for simplicity, we concatenate them together for processing in this branch. Specifically, still in layer $L_i$, assume we have a high-frequency token $T_i^H \in \mathbb{R}^{3m \times c}$. The token size of $T_i^H$ is tripled compared with the token size of $T_i^L$, as three high-frequency components are involved. Similar to the low-frequency branch, the high-frequency token $T_i^H$ is used to exploit the style information from the high-frequency components $f_i^{LH}$, $f_i^{HL}$ and $f_i^{HH}$, while at the same time following the low-rank adaptation paradigm (in Sec. 3.1).

Same as the low-frequency adaptation branch, we compute a similarity map $S_i^H \in \mathbb{R}^{n \times m}$ between the token $T_i^H$ and high-frequency components $f_i^{LH}$, $f_i^{HL}$ and $f_i^{HH}$, given by

$$S_i^H = \text{Softmax}(\frac{[f_i^{LH}, f_i^{HL}, f_i^{HH}] \times T_i^{H^\text{T}}}{\sqrt{c}}), \qquad (8)$$

where $[\cdot, \cdot]$ denotes the concatenation operation.

Then, the highlighted positions in $S_i^H$ reveal the predominant style responses from the source domain images. The high responses, which reflect more domain-specific styles, are supposed to be suppressed during training. It allows the fine-tuned VFM features to be less impacted by the domain-specific styles. Instance normalization [55, 31], which computes the channel-wise mean and standard deviation, is effective to eliminate the styles. To this end, an instance normalization is implemented on the feature-token similarity map $S_i^H$, given by

$$\widetilde{S}_i^H = \frac{S_i^H - \mu}{\sigma}, \quad \mu = \frac{1}{3m} \sum_{i=1}^{3m} S_i^H, \quad \sigma = \sqrt{\frac{1}{3m} \sum_{i=1}^{3m} (S_i^H - \mu)^2}. \qquad (9)$$

Then, we project the token feature $T_i^H$ into the high-frequency feature space by a multilayer perceptron (MLP) parameterized by weight parameters $W_i^3$ and bias parameters $b_i^3$, followed by the point-wise product with the similarity map $S_i^H$. The product with $S_i^H$ allows the token features $T_i^H$ to better align to the decoupled high-frequency features, which is less relevant to the source domain. Assume the output is denoted as $\overline{f}_i^H$. Briefly, this process can be mathematically expressed as

$$\overline{f}_i^H = \widetilde{S}_i^H \times [T_i^H \times W_i^3 + b_i^3]. \qquad (10)$$

Afterwards, we fuse the projected token features $\overline{f}_i^H$ with the high-frequency features $f_i^H$ by another MLP parameterized by weight parameters $W_i^4$ and bias parameters $b_i^4$, followed by the skip connection. Assuming the outputs of these three components are $\widetilde{f}_i^{LH}$, $\widetilde{f}_i^{HL}$ and $\widetilde{f}_i^{HH}$, this process can be mathematically computed as

$$[\widetilde{f}_i^{LH}, \widetilde{f}_i^{HL}, \widetilde{f}_i^{HH}] = [f_i^{LH}, f_i^{HL}, f_i^{HH}] + (\overline{f}_i^H + [f_i^{LH}, f_i^{HL}, f_i^{HH}]) \times W_i^4 + b_i^4. \qquad (11)$$

Finally, the low-frequency component $\widetilde{f}_i^{LL}$ and high-frequency components $\widetilde{f}_i^{LH}, \widetilde{f}_i^{HL}, \widetilde{f}_i^{HH}$ that have been processed by both branches are fused and transferred back by the inverse Haar wavelet transform. The output, denoted as $f_{i+1}$, is the input of the next frozen layer $L_{i+1}$.

## 4.4 Implementation Details

Same as the REIN [69] baseline, the loss function $\mathcal{L}$ of FADA directly inherits the losses from the Mask2Former decoder [12], given by

$$\mathcal{L} = \lambda_{ce}\mathcal{L}_{ce} + \lambda_{dice}\mathcal{L}_{dice} + \lambda_{cls}\mathcal{L}_{cls}, \qquad (12)$$

where $\mathcal{L}_{ce}$, $\mathcal{L}_{dice}$ and $\mathcal{L}_{cls}$ denote the cross-entropy loss, dice loss and classification loss. Here the hyper-parameters $\lambda_{ce}$, $\lambda_{dice}$ and $\mathcal{L}_{cls}$ are 5.0, 5.0 and 2.0, respectively.

By default we use DINO-V2 [53] as the frozen VFM, but the proposed FADA is also feasible to other VFMs. For fair evaluation, the Mask2Former segmentation decoder [13] is used to generate the pixel-wise prediction as REIN does. Same as the existing paradigm [69], the images are re-sized to $512 \times 512$ pixels before input to the models. The Adam optimizer with an initial learning rate of $1 \times 10^{-4}$ is used to train the model. The training process terminates after 20 epochs.

## 5 Experiments

### 5.1 Datasets & Evaluation Protocols

Five driving-scene semantic segmentation datasets that share 19 common scene categories are used for validation. Specifically, **CityScapes** (C) [16] consists of 2,975 and 500 images for training and validation, respectively. The images are captured under the clear conditions in tens of Germany cities. **BDD-100K** (B) [76] has 7,000 and 1,000 images for training and validation, respectively. The images are captured under diverse conditions from a variety of global cities. **Mapillary** (M) [51] is another large-scale semantic segmentation dataset, which consists of 25,000 images from diverse conditions. **SYNTHIA** (S) [64] is a synthetic driving-scene segmentation dataset, which has 9,400 images. **GTA5** (G) [63] is another synthetic dataset, which has 24,966 simulated images from the American street landscape.

Following the evaluation protocol of existing DGSS methods [55, 56, 14, 58], a certain dataset is used as the source domain for training and the rest four are used as unseen target domains for validation. Three commonly-used evaluation settings are: 1) G → C, B, M, S; 2) S → C, B, M, G; and 3) C → B, M, G, S. The evaluation metric is mean Intersection of Union (mIoU, in percentage %). All of our experiments are implemented and averaged by three independent repetitions, starting from different random seeds.

### 5.2 Comparison with State-of-the-art DGSS Methods

Existing DGSS methods are involved for comparison: 1) ResNet based methods, namely, IBN [55], IW [56], Iternorm [31], DRPC [78], ISW [15], GTR [57], DIRL [71], SHADE [82], SAW [58], WildNet [41], AdvStyle [83] and SPC [32]; 2) Mask2Former based methods, namely, HGFormer [19] and CMFormer [8]; 3) VFM based methods, namely, DIDEX [52] and REIN [69]. By default, the performance is directly cited from prior works [55, 56, 14, 58]. '-' denotes that the authors did not reported the results nor provided source code. '*' denotes re-implementation with official source code under all default settings.

**GTA5 Source Domain.** From left to right, the third column of Table 1 reports the performance. Compared with the VFM based REIN [69], the proposed FADA shows an mIoU improvement of 1.83%, 1.54%, 1.99% and 1.50% on the C, B, M and S target domains, respectively. In addition, the proposed FADA shows an average mIoU improvement of 20% and 10% when compared with ResNet and Mask2Former based DGSS methods, respectively.

**SYNTHIA Source Domain.** The fourth column of Table 1 shows that the proposed FADA achieves the state-of-the-art performance, outperforming the REIN by 1.45%, 1.41%, 1.22% and 1.29% mIoU on the C, B, M and G unseen target domains, respectively. In addition, the proposed FADA shows an average mIoU improvement of 15% and 6% over ResNet and Mask2Former based DGSS methods.

**CityScapes Source Domain.** The last column of Table 1 shows that the proposed FADA shows an mIoU improvement of 1.58%, 1.83%, 1.37% and 1.19% on the B, M, G and S unseen target domains, respectively. In addition, the proposed FADA shows an average mIoU improvement of 15% and 5% over existing ResNet and Mask2Former based methods, respectively.

### 5.3 Ablation Studies

**On Each Haar Component.** Five settings are involved for experiments: (1) No wavelet components are used. The model fine-tunes on the frozen VFM, which is a simplified version of REIN [69] removing the instance link module; (2) Only fine-tuning on the low-frequency component $f_i^{LL}$, and

Table 1: Performance comparison between the proposed FADA and existing DGSS methods. C: CityScapes [16]; B: BDD-100K [76]; M: Mapillary [51]; S: SYNTHIA [64]; G: GTA5 [63]. '-': results were not reported and official source code is not available; '*': only reported one decimal official results; '†': re-implementation with official source code under all default settings. Evaluation metric is mIoU in %. Top three results are highlighted as best, second and third, respectively.

| Method | Proc. & Year | Trained on GTA5 (G) | | | | Trained on SYNTHIA (S) | | | | Trained on Cityscapes (C) | | | |
|---|---|---|---|---|---|---|---|---|---|---|---|---|---|
| | | →C | →B | →M | →S | →C | →B | →M | →G | →B | →M | →G | →S |
| *ResNet based:* | | | | | | | | | | | | | |
| IBN [55] | ECCV2018 | 33.85 | 32.30 | 37.75 | 27.90 | 32.04 | 30.57 | 32.16 | 26.90 | 48.56 | 57.04 | 45.06 | 26.14 |
| IW [56] | CVPR2019 | 29.91 | 27.48 | 29.71 | 27.61 | 28.16 | 27.12 | 26.31 | 26.51 | 48.49 | 55.82 | 44.87 | 26.10 |
| Iternorm [31] | CVPR2019 | 31.81 | 32.70 | 33.88 | 27.07 | - | - | - | - | 49.23 | 56.26 | 45.73 | 25.98 |
| DRPC [78] | ICCV2019 | 37.42 | 32.14 | 34.12 | 28.06 | 35.65 | 31.53 | 32.74 | 28.75 | 49.86 | 56.34 | 45.62 | 26.58 |
| ISW [14] | CVPR2021 | 36.58 | 35.20 | 40.33 | 28.30 | 35.83 | 31.62 | 30.84 | 27.68 | 50.73 | 58.64 | 45.00 | 26.20 |
| GTR [57] | TIP2021 | 37.53 | 33.75 | 34.52 | 28.17 | 36.84 | 32.02 | 32.89 | 28.02 | 50.75 | 57.16 | 45.79 | 26.47 |
| DIRL [71] | AAAI2022 | 41.04 | 39.15 | 41.60 | - | - | - | - | - | 51.80 | - | 46.52 | 26.50 |
| SHADE [82] | ECCV2022 | 44.65 | 39.28 | 43.34 | - | - | - | - | - | 50.95 | 60.67 | 48.61 | 27.62 |
| SAW [58] | CVPR2022 | 39.75 | 37.34 | 41.86 | 30.79 | 38.92 | 35.24 | 34.52 | 29.16 | 52.95 | 59.81 | 47.28 | 28.32 |
| WildNet [41] | CVPR2022 | 44.62 | 38.42 | 46.09 | 31.34 | - | - | - | - | 50.94 | 58.79 | 47.01 | 27.95 |
| AdvStyle [83] | NeurIPS2022 | 39.62 | 35.54 | 37.00 | - | 37.59 | 27.45 | 31.76 | - | - | - | - | - |
| SPC [32] | CVPR2023 | 44.10 | 40.46 | 45.51 | - | - | - | - | - | - | - | - | - |
| BlindNet [2] | CVPR2024 | 45.72 | 41.32 | 47.08 | 31.39 | - | - | - | - | 51.84 | 60.18 | 47.97 | 28.51 |
| *Mask2Former:* | | | | | | | | | | | | | |
| HGFormer*[19] | CVPR2023 | - | - | - | - | - | - | - | - | 53.4 | 66.9 | 51.3 | 33.6 |
| CMFormer [8] | AAAI2024 | 55.31 | 49.91 | 60.09 | 43.80 | 44.59 | 33.44 | 43.25 | 40.65 | 59.27 | 71.10 | 58.11 | 40.43 |
| *VFM based:* | | | | | | | | | | | | | |
| DIDEX*[52] | WACV2024 | 62.0 | 54.3 | 63.0 | - | - | - | - | - | - | - | - | - |
| REIN* [69] | CVPR2024 | 66.4 | 60.4 | 66.1 | 48.86† | 48.59† | 44.42† | 48.64† | 46.97† | 63.54† | 74.03† | 62.41† | 48.56† |
| FADA (Ours) | - | **68.23** | **61.94** | **68.09** | **50.36** | **50.04** | **45.83** | **49.86** | **48.26** | **65.12** | **75.86** | **63.78** | **49.75** |
| | | ↑1.83 | ↑1.54 | ↑1.99 | ↑1.50 | ↑1.45 | ↑1.41 | ↑1.22 | ↑1.29 | ↑1.58 | ↑1.83 | ↑1.37 | ↑1.19 |

Table 2: Ablation studies on each component of the proposed FADA. $LL$, $LH$, $HL$ and $HH$ denote the $f_i^{LL}$, $f_i^{LH}$, $f_i^{HL}$ and $f_i^{HH}$ components, respectively. ✓ refers to that fine-tuning is implemented. Evaluation metric is mIoU in %.

| Frequency Components | | | | Trained on CityScapes (C) | | | | Trained on SYNTHIA (S) | | | |
|---|---|---|---|---|---|---|---|---|---|---|---|
| $LL$ | $LH$ | $HL$ | $HH$ | →B | →M | →G | →S | →C | →B | →M | →G |
| ✗ | ✗ | ✗ | ✗ | 62.43 | 73.05 | 61.29 | 47.61 | 48.03 | 43.27 | 47.85 | 46.02 |
| ✓ | ✗ | ✗ | ✗ | 63.85 | 74.16 | 62.04 | 48.68 | 48.79 | 44.81 | 48.96 | 47.35 |
| ✓ | ✓ | ✗ | ✗ | 64.04 | 74.89 | 62.95 | 48.92 | 49.18 | 45.07 | 49.13 | 48.07 |
| ✓ | ✓ | ✓ | ✗ | 64.69 | 75.16 | 63.20 | 49.35 | 49.62 | 45.37 | 49.50 | 48.16 |
| ✓ | ✓ | ✓ | ✓ | **65.12** | **75.86** | **63.78** | **49.75** | **50.04** | **45.83** | **49.86** | **48.26** |

Table 3: Ablation studies of the rank $r$ on generalization performance. Evaluation metric is mIoU in %.

| Method | Trained on Cityscapes (C) | | | |
|---|---|---|---|---|
| | →B | →M | →G | →S |
| 4 | 64.21 | 74.96 | 62.79 | 48.68 |
| 8 | 64.73 | 75.18 | 63.06 | 49.03 |
| 16 | 65.12 | **75.86** | **63.78** | **49.75** |
| 32 | **65.28** | 75.34 | 63.56 | 49.42 |
| 64 | 64.85 | 75.12 | 62.38 | 49.64 |

Table 4: Generalization ability test of the proposed FADA on different VFM models. One decimal result is reported and compared following prior references.

| Backbone | Fine-tune Method | Trainable Params* | mIoU | | | |
|---|---|---|---|---|---|---|
| | | | Citys | BDD | Map | Avg. |
| | Full | 304.15M | 51.3 | 47.6 | 54.3 | 51.1 |
| CLIP [62] | Freeze | 0.00M | 53.7 | 48.7 | 55.0 | 52.4 |
| | REIN [69] | 2.99M | 57.1 | 54.7 | 60.5 | 57.4 |
| | FADA | 11.65M | **58.7** | **55.8** | **62.1** | **58.9** |
| | Full | 632.18M | 57.6 | 51.7 | 61.5 | 56.9 |
| SAM [39] | Freeze | 0.00M | 57.0 | 47.1 | 58.4 | 54.2 |
| | REIN [69] | 4.51M | 59.6 | 52.0 | 62.1 | 57.9 |
| | FADA | 16.59M | **61.0** | **53.2** | **63.4** | **60.0** |
| | Full | 304.24M | 62.1 | 56.2 | 64.6 | 60.9 |
| EVA02 [20] | Freeze | 0.00M | 56.5 | 53.6 | 58.6 | 56.2 |
| | REIN [69] | 2.99M | 65.3 | 56.3 | 63.6 | 63.6 |
| | FADA | 11.65M | **66.7** | **61.9** | **66.1** | **64.9** |
| | Full | 304.20M | 63.7 | 57.4 | 64.2 | 61.7 |
| DINOV2 [53] | Freeze | 0.00M | 63.3 | 56.1 | 63.9 | 61.1 |
| | REIN [69] | 2.99M | 66.4 | 60.4 | 66.1 | 64.3 |
| | FADA | 11.65M | **68.2** | **62.0** | **68.1** | **66.1** |

Table 5: Generalization performance comparison on the four adverse condition domains from ACDC dataset [65]. CityScapes as the source domain. Top three results are highlighted as best, second and third, respectively.

| Method | Trained on Cityscapes (C) | | | | |
|---|---|---|---|---|---|
| | →Fog | →Night | →Rain | →Snow | mean |
| *ResNet Based:* | | | | | |
| IBN [55] | 63.8 | 21.2 | 50.4 | 49.6 | 43.7 |
| Iternorm [30] | 63.3 | 23.8 | 50.1 | 49.9 | 45.3 |
| IW [56] | 62.4 | 21.8 | 52.4 | 47.6 | 46.6 |
| ISW [14] | 64.3 | 24.3 | 56.0 | 49.8 | 48.1 |
| *Transformer Based:* | | | | | |
| ISSA [45] | 67.5 | 33.2 | 55.9 | 53.2 | 52.5 |
| HGFormer [19] | 69.9 | 52.7 | 72.0 | 68.6 | 67.2 |
| Mask2Former [13] | 73.4 | 37.1 | 63.6 | 62.5 | 58.0 |
| CMFormer [8] | 77.8 | 33.7 | 67.6 | 64.3 | 60.9 |
| *VFM based:* | | | | | |
| REIN† [69] | 79.5 | 55.9 | 72.5 | 70.6 | 69.6 |
| **Ours** | **80.2** | **57.4** | **75.0** | **73.5** | **71.5** |
| | ↑0.7 | ↑1.5 | ↑2.5 | ↑2.9 | ↑1.9 |

directly fusing with three high-frequency components without fine-tuning; (3) fine-tuning on $f_i^{LL}$ and $f_i^{LH}$; (4) fine-tuning on $f_i^{LL}$, $f_i^{LH}$ and $f_i^{HL}$; (5) fine-tuning on all the four frequency components (Ours). Table 2 reports the results. Harnessing the low-frequency features leads to an average of 1% mIoU improvement on most experimental settings. In addition, fine-tuning on each high-frequency component leads to a slight improvement on unseen target domains. It demonstrates the necessity to decouple the impact of cross-domain styles.

**On Low-Rank Dimension $r$.** By default we set the low-rank dimension $r$ as 16. We further test the generalization performance when $r$ is 4, 8, 32 and 64, respectively. Table 3 reports the performance when CityScapes is used as the source domain. When $r$ is 16 or 32, the performance on unseen target domains shows the most stable performance. However, when $r$ is too small (*e.g.*, 4 or 8) or too large

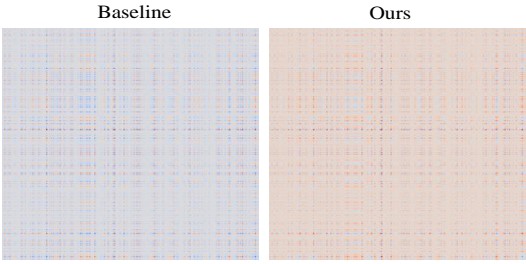

| Baseline | Ours |
|---|---|

Figure 3: Channel-wise correlation matrix of the features from last VFM layer between source domain (C) and unseen domain (B). The brighter a cell is, the higher response.

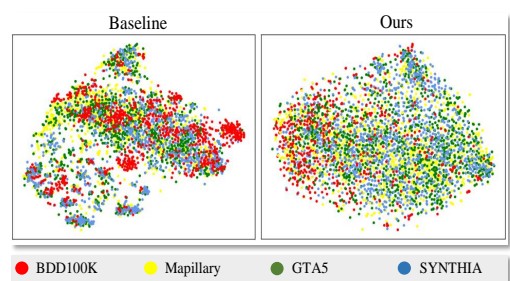

| Baseline | Ours |
|---|---|

● BDD100K ● Mapillary ● GTA5 ● SYNTHIA

Figure 4: t-SNE visualization. Feature embedding is extracted from the last VFM layer. Left: baseline; Right: ours.

(*e.g.*, 64), the performance on unseen target domains demonstrates a clear decline, which can be explained by the under-fitting and over-fitting, respectively.

**Understanding from Cross-Domain Feature Correlation.** We display the channel-wise correlation matrix of the last-layer feature embeddings from C source domain and B target domain. The results are displayed in Fig. 3. Brighter indicates higher response. FADA allows both low- and high-frequency token features from the source domain and unseen target domains to show similar channel-wise activation response, which allows the model to be better generalized to unseen target domains.

**T-SNE Visualization.** We display the features before the segmentation decoder by t-SNE visualization. The experiments are conducted under the C → B, M, G, S setting. The feature space of the original REIN and the proposed FADA is visualized in Fig. 4. The samples from different unseen target domains are more uniformly distributed by the proposed FADA, narrowing the domain gap.

**Understanding the Benefit of Instance Normalization to Mitigate Domain-specific Information.**
We extract the three high-frequency components from the last VFM layer, and display them by t-SNE visualization. The feature space without (denoted as w.o.) and with (denoted as with) implementing the instance normalization function is visualized in the first and second row of Fig. 5, respectively. The experiments are conducted under the C → B, M, G, S setting. It is observed that the implementation of instance normalization allows the samples from different unseen target domains to be more uniformly distributed, indicating its effectiveness to mitigate the domain-specific information containing in the high-frequency components.

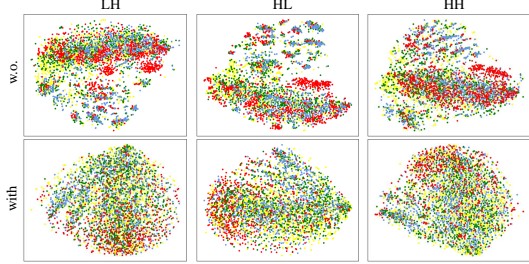

Figure 5: Impact of instance normalization on the domain generalization property of high-frequency components. w.o./with: without/with implementing instance normalization.

## 5.4 Generalization on Other Settings

**To Different VFMs.** We test the translation ability of the proposed FADA to other VFMs, namely, CLIP [62], SAM [39] and EVA02 [20]. For comprehensive evaluation, each VFM is validated under full-training, fine-tuning (REIN), or frozen scheme [69], respectively. The reported one decimal results are directly cited from [69]. Experiments are conducted under the G → {C, B, M} setting. Table 4 shows the superiority of FADA when embedded into these VFMs. For comparison with parameter-efficient fine-tuning (PEFT) methods, please refer to Table 7 in the supplementary material.

**To Adverse Domains** Adverse Conditions Dataset with Correspondence (ACDC) [65] is a semantic segmentation dataset that consists of samples from four types of adverse conditions, namely, rain, fog, night and snow. Table 5 shows that the proposed FADA outperforms existing DGSS methods by up to 0.7%, 1.5%, 2.5% and 2.9% on the fog, night, rain and snow domains, respectively.

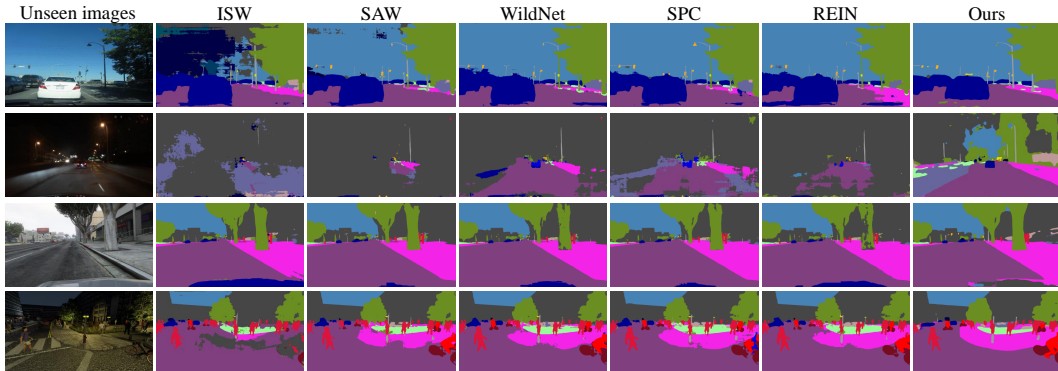

Figure 6: Exemplar segmentation results of existing DGSS methods (ISW [14], SAW [58], WildNet [41], SPC [32], CMFormer [8], and REIN [69]) and FADA under the C →{B, G, M, S} setting.

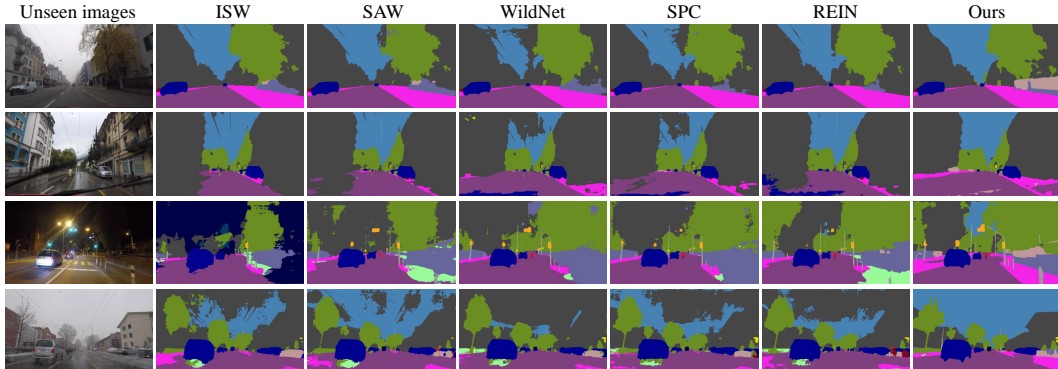

Figure 7: Exemplar segmentation results of existing DGSS methods (ISW [14], SAW [58], WildNet [41], SPC [32], CMFormer [8], and REIN [69]) and FADA under the C → four ACDC setting.

### 5.5 Quantitative Segmentation Results

Some exemplar segmentation results are compared under the C → B, M, G, S and C → adverse domains are provided in Fig. 6 and Fig. 7, respectively. The proposed FADA shows better pixel-wise prediction than not only ResNet based methods (*i.e.*, ISW [14], SAW [58], WildNet [41], and SPC [32]) and Mask2Former based methods (*i.e.*, CMFormer [8]), but also VFM based REIN [69].

## 6 Conclusion

In this paper, we focused on adapting VFM for DGSS by exploiting the style-invariant properties from the VFMs, and presented a novel **F**requency-**ADA**pted learning scheme to push this frontier. Concisely, Haar wavelet transform was introduced to decouple the frozen VFM features into low- and high-frequency components, which contain more scene content and style information, respectively. We innovatively modified the low-rank adaptation paradigm to both frequency features, and alleviated the impact of cross-domain variation on high-frequency features. Consequently, the model achieved a better generalization on unseen target domains. Extensive experiments and ablation studies on a variety of settings showed the effectiveness of the proposed FADA.

**Limitation Discussion & Broader Societal Impact.** The proposed FADA handles the low- and high-frequency features separately, which increases the trainable parameters compared with prior work (Table 4). However, the increase of about 6M parameters is acceptable. The proposed FADA advances the reliability and safety of autonomous driving and alleviates human involvement, benefiting the human well-being. We do not envision any negative social impact.

**Acknowledgments and Disclosure of Funding.** This work was supported by the Science and Technology Major Project of Guangxi (AA22096030 and AA22096032), and National Key R&D Program of China under Grant (2020AAA0109500 and 2020AAA0109501).

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

# A  Appendix / supplemental material

## A.1  Theoretical Analysis on Haar Wavelets

The Haar transform [60] cross-multiplies a function with various shifts and stretches, which has been demonstrated to be effective in applications, such as signal and image processing. In other words, it is able to analyze the local aspects of a signal.

**Definition 1.  Haar Scaling Function.** *Given an input signal $x$, the Haar scaling function is mathematically defined as*

$$\phi(t) = \begin{cases} 1 & 0 \le t < 1 \\ 0 & \text{otherwise} \end{cases}. \tag{13}$$

Given the space of all functions of the form $\sum_{k \in \mathbb{Z}} a_k \phi(x - k)$ as $V_0$, where $k \in \mathbb{Z}$ is an arbitrary integer, and $a_k \in \mathbb{R}$. As each element of $V_0$ is zero outside a bounded set, such a function $a_k \phi(x - k)$ has *finite or compact support*.

**Definition 2. Basis of the Step Function Space.** *Given an arbitrary nonnegative integer $j \in \mathbb{Z}_0^+$, Let $V_j$ denote the step function space at the level $j$, which is spanned by the set*

$$\{\cdots, \phi(2^j x + 1), \phi(2^j x), \phi(2^j x - 1), \cdots\}. \tag{14}$$

**Theorem 1.** *(1) A function $f(x)$ belongs to $V_0 \iff f(2^j x)$ belongs to $V_j$. (2) A function $f(x)$ belongs to $V_j \iff f(2^{-j} x)$ belongs to $V_0$.*

**Proof.** (1) If $f(x) \in V_0$, then we have $f(x) = \sum_{k \in \mathbb{Z}} a_k \phi(x - k)$, where $a_k \in \mathbb{R}$. Then, $f(2^j x) = \sum_{k \in \mathbb{Z}} a_k \phi(2^j x - k)$. It means $f(2^j x) \in V_j$. (2) The proof of (2) is similar.

**Theorem 2.** *The set of functions $\{2^{j/2} \phi(2^j x - k), k \in \mathbb{Z}\}$ is an orthonormal basis of $V_j$.*

**Proof.** The norm of a certain basis $2^{j/2} \phi(2^j x - k)$ is $|2^{j/2} \phi(2^j x - k)| = 2^{j/2} |\phi(2^j x - k)| = 2^{j/2} \cdot \frac{1}{2^{j/2}}$ =1. (2) For any two basis $m$ and $n$ ($m \ne n$), $< 2^{m/2} \phi(2^m x - k), 2^{n/2} \phi(2^n x - k) > = 2^{m/2} \cdot 2^{n/2} \cdot < \phi(2^m x - k), \phi(2^n x - k) > = 0$. Thus, the set of functions $\{2^{j/2} \phi(2^j x - k), k \in \mathbb{Z}\}$ is an orthonormal basis of $V_j$.

**Definition 3. Haar Wavelet Function.** *The Haar wavelet is the function $\psi(x) = \phi(2x) - \phi(2x - 1)$.*

**Theorem 3.** *The Haar wavelet function $\psi(x) \in V_1$, and is orthogonal to $V_0$.*

**Lemma 1.** *Any function $f_1(x) = \sum_{k \in \mathbb{Z}} a_k \phi(2x - k) \in V_1$, i.e., orthogonal to each $\phi(x - l), l \in \mathbb{Z}$ if and only if $a_1 = -a_0$, $a_3 = -a_2$, $\cdots$*

**Proof.** Given $\phi(x) \in V_0$, if $f_1(x) = \sum_{k \in \mathbb{Z}} a_k \phi(2x - k) \perp V_0$, then $\sum_{k \in \mathbb{Z}} a_k \phi(2x - k) \perp \phi(x)$. Therefore, $f_1(x) \perp V_0$ if and only if $< \sum_{k \in \mathbb{Z}} a_k \phi(2x - k), \phi(x) >= 0$. As $\phi(x) = \phi(2x) - \phi(2x - 1)$, we have

$$< \sum_{k \in \mathbb{Z}} a_k \phi(2x - k), \phi(2x) - \phi(2x - 1) >= a_0 < \phi(2x), \phi(2x) > + a_1 < \phi(2x - 1), \phi(2x - 1) >= 0. \tag{15}$$

Thus, $a_0 = -a_1$. Similarly, by inspecting $< \sum_{k \in \mathbb{Z}} a_k \phi(2x - k), \phi(x - 1) >= 0$, we have $a_2 + a_3 = 0, \cdots$.

**Proof of Theorem 3.** Based on *Lemma 1*, we have

$$f_1(x) = \sum_{k \in \mathbb{Z}} a_{2k} (\phi(2x - k) - \phi(2x - k - 1)) = \sum_{k \in \mathbb{Z}} a_{2k} \psi(x - k). \tag{16}$$

## A.2  Impact on Token Length $m$

By default we set the token length $m$ in the proposed FADA as 100. To study its impact on the unseen target domain, the scenarios when $m$ are set 25, 50, 75, 125, 150 and 175 are tested. CityScapes is used as the source domain. The results on B, M, G and S unseen target domains are displayed in Fig. 8 a, b, c and d, respectively. It is observed that, the generalization performance when $m$ ranges from 75 to 125 is relatively stable, while the performance when $m$ is too small (*i.e.*, 25, 50) or too large (*i.e.*, 150, 175) shows a slight decline.

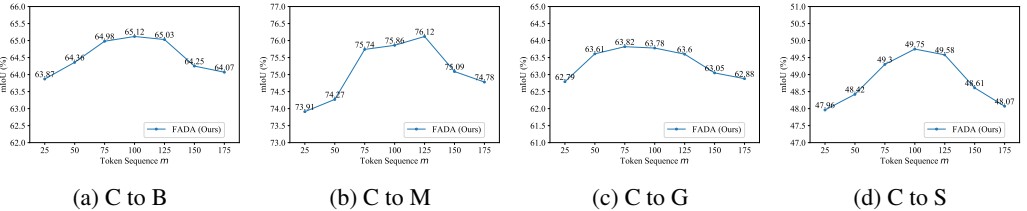

| (a) C to B | (b) C to M | (c) C to G | (d) C to S |

Figure 8: Ablation studies on the token length $m$. Experiments are conducted under the C [16] → B [76], M [51], G [63], S [64] settings, which are displayed from left to right.

Table 6: Ablation study on the positions of the frequency adapters. GTA5 as the source domain. CityScapes, BDD and Mappilary are unseen target domains. Evaluation metric is mIoU in %. Top three results are highlighted as best , second and third , respectively.

| Method | Citys | BDD | Map | Avg. |
|---|---|---|---|---|
| Full | 63.7 | 57.4 | 64.2 | 61.7 |
| Freeze | 63.3 | 56.1 | 63.9 | 61.1 |
| REIN [69] | 66.4 | 60.4 | 66.1 | 64.3 |
| Shallow | 67.6 | 61.5 | 67.4 | 65.5 |
| Deep | 67.3 | 61.2 | 67.0 | 65.2 |
| FADA | **66.7** | **61.9** | **66.1** | **64.9** |

## A.3 Impact on the Position of the Frequency Adapter

The high-level idea of our intuition focuses on the frequency space. As existing LoRA based and side-adapter based methods implement the adaptation on each of the transformer layer, therefore we treat the frequency space as a whole, and embed it into each transformer layer.

Nevertheless, it would be meaningful to have an in-depth analysis on the learning behaviour from shallow to deep. To this end, apart from the REIN baseline [69] and the proposed FADA, we further provide two experiments, where the frequency adapter is attached in the first half seven layers (denoted as shallow) and the second half seven layers (denoted as deep), respectively. Results in Table 6 show that:

- Using the frequency adapter on the first half layers (shallow) shows a slightly better performance than on the second half layers (deep). It may be explained that the shallower features contain more cross-domain styles, such as illumination, landscape and *etc*.

- Using the frequency adapter on all layers (ours) achieves the best performance, indicating its effectiveness on all layers.

## A.4 Comparison with Token Fine-Tuning Methods

We further compare the proposed method with some existing parameter-efficient fine-tuning (PEFT) methods, namely, AdvStyle [83], PASTA [9], GTR-LTR [57], LoRA [28], AdaptFormer [10], VPT [38] and REIN [69]. Following the setting in REIN [69], GTAV is used as the source domain. BDD, CityScapes and Map are used as unseen target domains. Results in Table 7 show that the proposed FADA outperforms these methods on all unseen target domains.

## A.5 More Visual Results

Fig. 9 and Fig. 10 show more results under C → B, M, G, S and C → ACDC setting. On both settings, the segmentation results show that the proposed FADA shows better pixel-wise prediction than the compared DGSS methods, especially in terms of the completeness of objects.

Table 7: Performance Comparison of the proposed FADA against other DGSS and PEFT methods under the G→ C, B, M setting. The best results are highlighted. ∗ denotes trainable parameters in backbones. Top three results are highlighted as best , second and third , respectively.

| Backbone | Fine-tune Method | Trainable Params* | mIoU | | | |
|---|---|---|---|---|---|---|
| | | | Citys | BDD | Map | Avg. |
| EVA02 [20] | Full | 304.24M | 62.1 | 56.2 | 64.6 | 60.9 |
| | +AdvStyle [83] | 304.24M | 63.1 | 56.4 | 64.0 | 61.2 |
| | +PASTA [9] | 304.24M | 61.8 | 57.1 | 63.6 | 60.8 |
| | Freeze | 0.00M | 56.5 | 53.6 | 58.6 | 56.2 |
| | +AdvStyle [83] | 0.00M | 51.4 | 51.6 | 56.5 | 53.2 |
| | +PASTA [9] | 0.00M | 57.8 | 52.3 | 58.5 | 56.2 |
| | +GTR-LTR [57] | 0.00M | 52.5 | 52.8 | 57.1 | 54.1 |
| | +LoRA [28] | 1.18M | 55.5 | 52.7 | 58.3 | 55.5 |
| | +AdaptFormer [10] | 3.17M | 63.7 | 59.9 | 64.2 | 62.6 |
| | +VPT [38] | 3.69M | 62.2 | 57.7 | 62.5 | 60.8 |
| | +Rein (ours) | 2.99M | 65.3 | 60.5 | 64.9 | 63.6 |
| | +FADA | 11.65M | **66.7** | **61.9** | **66.1** | **64.9** |
| DINOv2 (Large) [53] | Full | 304.20M | 63.7 | 57.4 | 64.2 | 61.7 |
| | +AdvStyle [83] | 304.20M | 60.8 | 58.0 | 62.5 | 60.4 |
| | +PASTA [9] | 304.20M | 62.5 | 57.2 | 64.7 | 61.5 |
| | +GTR-LTR [9] | 304.20M | 62.7 | 57.4 | 64.5 | 61.6 |
| | Freeze | 0.00M | 63.3 | 56.1 | 63.9 | 61.1 |
| | +AdvStyle [83] | 0.00M | 61.5 | 55.1 | 63.9 | 60.1 |
| | +PASTA [9] | 0.00M | 62.1 | 57.2 | 64.5 | 61.3 |
| | +GTR-LTR [9] | 0.00M | 60.2 | 57.7 | 62.2 | 60.0 |
| | +LoRA [28] | 0.79M | 65.2 | 58.3 | 64.6 | 62.7 |
| | +AdaptFormer [10] | 3.17M | 64.9 | 59.0 | 64.2 | 62.7 |
| | +VPT [38] | 3.69M | 65.2 | 59.4 | 65.5 | 63.3 |
| | +WHT [73] | 3.51M | 65.8 | 58.9 | 65.3 | 63.3 |
| | +FourierFT [21] | 0.67M | 66.1 | 59.2 | 65.8 | 63.7 |
| | +REIN [69] | 2.99M | 66.4 | 60.4 | 66.1 | 64.3 |
| | +FADA | 11.65M | **68.2** | **62.0** | **68.1** | **66.1** |

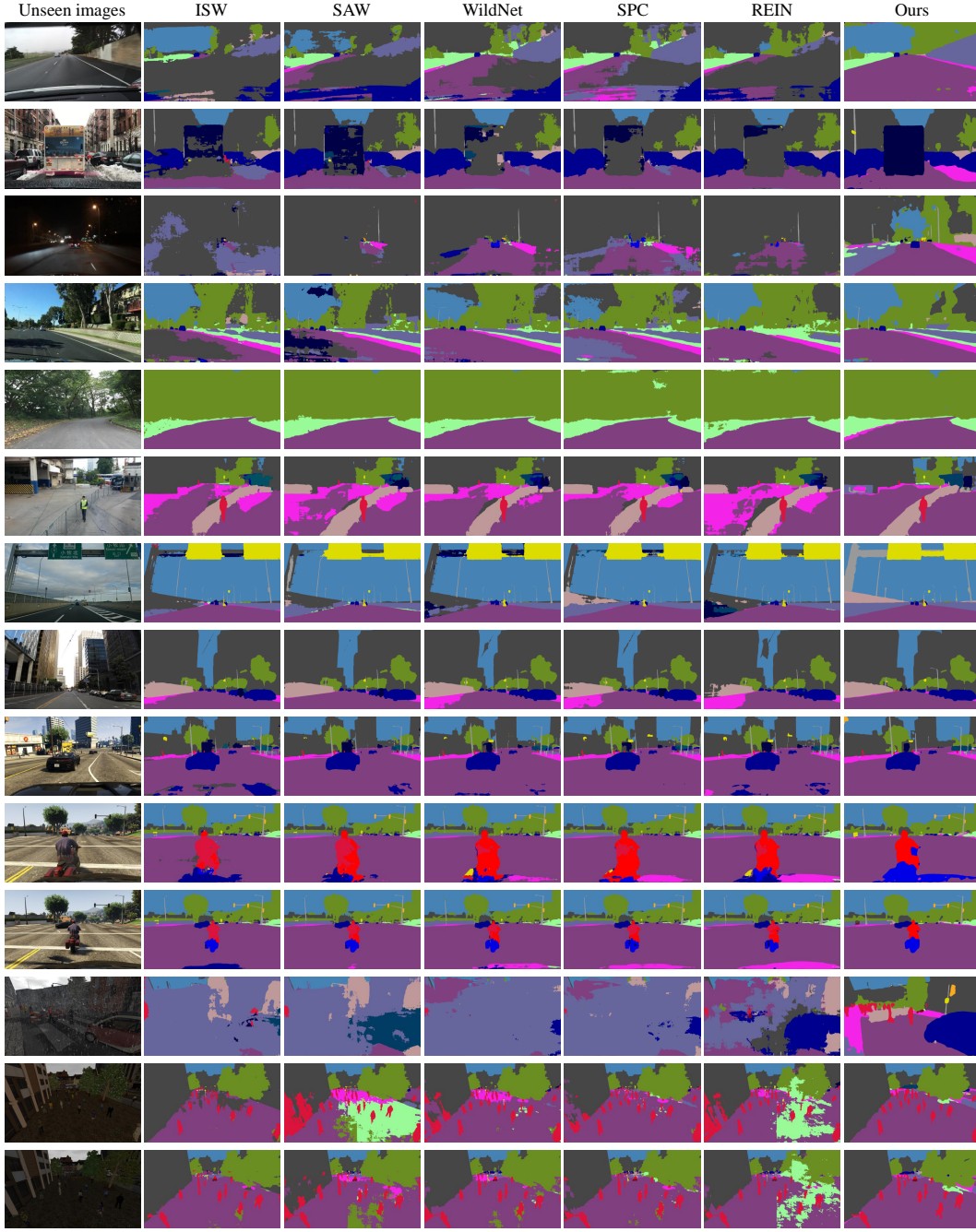

Figure 9: Visual segmentation results on unseen target domains under the C → B, M, G, S setting. The proposed FADA is compared with ISW [15], SAW [58], WildNet [41], SPC [32] and Rein [**?** ].

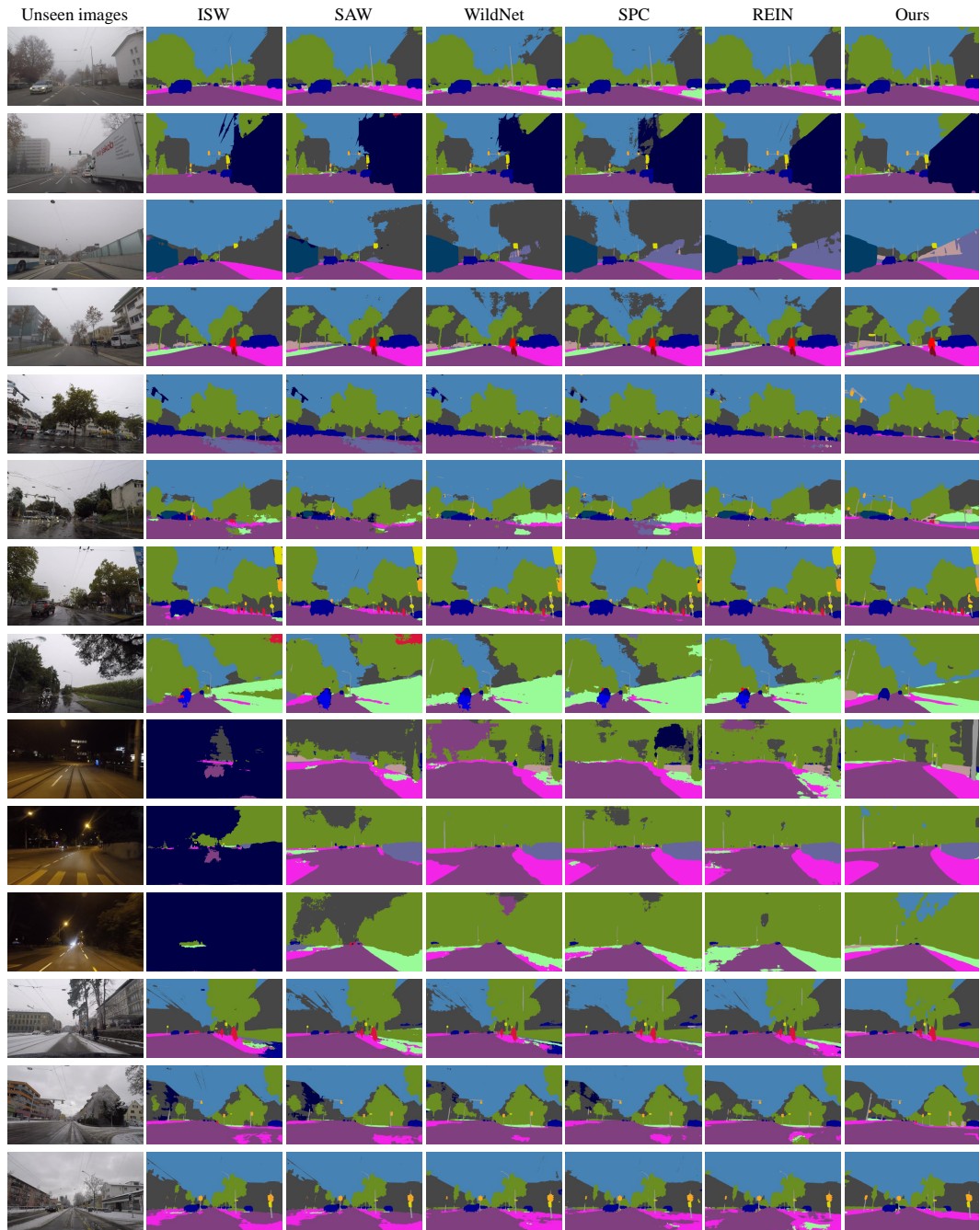

Figure 10: Visual segmentation results on unseen target domains under the C → ACDC setting. The proposed FADA is compared with ISW [15], SAW [58], WildNet [41], SPC [32] and Rein [? ].

