# OpenReview forum: "Learning Frequency-Adapted Vision Foundation Model for Domain Generalized Semantic Segmentation"
_NeurIPS.cc/2024/Conference — NeurIPS 2024 poster_

### Official Review · Reviewer_ojsk · 2024-07-02

**Soundness:** 3
**Presentation:** 3
**Contribution:** 3
**Rating:** 5
**Confidence:** 2

**Summary:**

This paper deals with adapting a computer vision foundational model to a new domain.  They propose to use the frequency based on Haar wavelets to  to decouple the style and content information and then to address separately  content and style  domain adaptation. Experiments show state of the art results.

**Strengths:**

The paper is relatively well-written. Adapting high and low frequency components separately seems like a good idea although it could be more motivated.

Figure 1 is compelling.

Experimental results show consistent improvement.

Tested across various foundational models.

**Weaknesses:**

The adaptation for low-frequency and high-frequency components are different. The paper could provide more intuition to explain the particular adaptations chosen for low and high frequency components.

**Questions:**

see above

**Limitations:**

Yes

---

> ### Author Rebuttal · Authors · 2024-08-06
>
> **Q1**: The adaptation for low-frequency and high-frequency components are different. The paper could provide more intuition to explain the particular adaptations chosen for low and high frequency components.
>
> **R**:
> Thanks for your general positive comments on our work and your insightful comment, so that we could have the chance to provide a more in-depth analysis on the particular adaptations.
>
> In general, the adaptation on high-frequency components is further implemented by an instance normalization when compared to the low-frequency components.
> Instance normalization, which computes the channel-wise mean and standard deviation, is effective to eliminate the domain-specific styles [44,27].
>
> To further validate this point, in the attached file in general response, we visualize the t-SNE feature space of the samples from four unseen domains.
> The results without and with implementing the instance normalization on the high-frequency components are displayed in the first and second row of Fig.R1, respectively.
> Fig.R1 is attached in the 1-pg PDF in the general response.
> Notice that, when without instance normalization, the method is the same as using the same adaptation on the low-frequency component.
>
> It can be seen that, the use of instance normalization alleviates the existence of the domain-specific clusters (in the same color) and allows samples from different domains to be more uniformly distributed (in different color).
> The domain-specific clusters indicate that samples from the same domain have less distance and higher similarity in the feature space.
> It indicates that the different adaptation for high-frequency component is more effective to develop the style-invariant property.
>
> Finally, should you have further suggestions and questions, we are glad to address during the discussion stage.

---

### Official Review · Reviewer_4xQk · 2024-07-10

**Soundness:** 2
**Presentation:** 1
**Contribution:** 2
**Rating:** 3
**Confidence:** 5

**Summary:**

The paper proposes a Frequency-Adapted (FADA) learning scheme, where Haar wavelet transformation is introduced to decouple the frozen VFM features into low- and high-frequency components. Experiments demonstrate the proposed method achieves a better generalization on unseen target domains.

**Strengths:**

The proposed method achieves good performance.

**Weaknesses:**

1. The writing and logic are poor. For example, the relationship among the existing three DGSS category methods is missing. Why explore the style-invariant properties of VFM is important and urgent? The introduction is not accompanied by an explanation of Figure 1.
2. What are the advantages of 1) leveraging Haar wavelet for domain generalization and 2) enhancing the generalization ability of VFM features via frequency space in Domain Generalization by Frequency Decoupling over existing related works?
3. [1] also focuses on mitigating the effects of style variations on the DGSS. What is the difference between the paper with [1]?
[1] Style Blind Domain Generalized Semantic Segmentation via Covariance Alignment and Semantic Consistence Contrastive Learning. CVPR 2024

**Questions:**

I'm more concerned about [1] also focusing on mitigating the effects of style variations on the DGSS. What is the difference between the paper with [1]? Besides, why is exploring the style-invariant properties of VFM important and urgent? What are the advantages of 1) leveraging Haar wavelet for domain generalization and 2) enhancing the generalization ability of VFM features via frequency space in Domain Generalization by Frequency Decoupling over existing related works?
[1] Style Blind Domain Generalized Semantic Segmentation via Covariance Alignment and Semantic Consistence Contrastive Learning. CVPR 2024

**Limitations:**

The proposed method heavily relies on Low-/High-Frequency Decomposition. However, the manner is fixed and not adjusted dynamically.

---

> ### Author Rebuttal · Authors · 2024-08-06
>
> **Q1**: Writing \& logic. 1) the relationship among three DGSS category methods; 2) Why the style-invariant properties of VFM is important and urgent? 3) The introduction is not accompanied by an explanation of Fig. 1.
>
> **R**: Thanks for your valuable feedback so that we could have the chance to introduce a native speaker to thoroughly polish the writing and logic. The mentioned problems are clarified as follows and will be incorporated accordingly.
>
> 1) Existing three types of DGSS methods, either decouple the style, augment the style or stabilize the content, are all full-training paradigm (on a CNN or Vision Transformer encoder).
> In contrast, the realm of VFM provides a new paradigm for DGSS, which only fine-tunes on a minor part of model parameters instead of full-training.
> The motivation of this work is to allow the VFM features to stabilize the content while at the same time be invariant to the style shift, which aligns the advantages of different types of existing DGSS methods.
>
> 2) An ideal representation for DGSS is to learn stable content representation despite the cross-domain style shift [44,26,13,45,63,60,47].
> However, the fine-tuning of VFM is highly dependent on the low-dimensional intrinsic embeddings [58], which is subtle and fragile to the distribution shift, which is posed by the style shift in the context of DGSS.
> Therefore, an ideal VFM representation for DGSS is supposed to demonstrate robust style-invariant property, so that it is able to demonstrate robust scene representation from different domains.
>
> 3) We would like to kindly raise the reviewer's attention that, the explanation of Fig.1 is in the third line of the introduction; while the caption of Fig.1 further details the the domain generalization property of the high-/low- frequency components of Haar wavelet transformation.
>
> **Q2**: Advantages of 1) Haar wavelet for domain generalization, 2) enhancing generalization of VFM features by Frequency Decoupling over existing related works?
>
> **R**:
> Thanks for your constructive comments. To clarify:
>
> 1) Both FFT based methods and Haar wavelet can decouple a scene representation into low- and high- frequency components.
> However, compared with FFT, Haar wavelet transformation consists of orthonormal basis, and warrants the orthonormality between the low- and high- frequency components.
> Orthonormality is essentially a type of de-correlation, which provides a better separation between the low- and high- frequency components.
> In the context of DGSS, the orthonormality of Haar wavelet transformation allows a better seperation between the scene content and the style.
> Meanwhile, from the experimental side, Table R1 (in general response) shows that our Haar wavelet based method shows a superior performance on all unseen target domains than the FFT based FourierFT [a].
> This outcome further demonstrates the advantage of Haar wavelet over existing Fourier transformation based methods.
>
> 2) We would like to kindly raise the reviewer's attention that, harnessing VFM for DGSS is an emerging research line.
> By the time of submission, the closest work is REIN [58].
> REIN [58] directly implements low-rank adaptation to learn the entire scene representation. It does not decouple the content and style, which is necessary in visual domain generalization.
> In contrast, the advantage of the proposed method is that, it provides a feasible path to decouple the style from the content by the separation between high- and low- frequency components.
> This decoupling allows the representation learning to handle the style and content in a divide-and-conquer manner, where the style representation is supposed to be invariant to the domain, while the scene content is supposed to be stable.
>
> 3) The research line of frequency decoupling based methods is focused on generic domain generalization (classification task), not for DGSS or VFM based DGSS.
>
> **Q3**: What is the difference between the paper with [1]?
>
> [1] Style Blind Domain Generalized Semantic Segmentation via Covariance Alignment and Semantic Consistence Contrastive Learning. CVPR 2024.
>
> **R**:
> Thanks for your reference suggestion, so that we could have a chance to discuss the difference between the recent CVPR 2024 work BlindNet.
> We would like to kindly raise the reviewer's attention that, this paper is essentially different from the proposed method in many critical aspects.
>
> 1) Completely different methodology design. BlindNet first augments the style diversity from the image, and constraints the scene representation by designing style similarity and content consistency based losses.
> In contrast, the proposed method decouples VFM features in the frequency space by Haar wavelet to decouple the style and content separately.
> It does not reply on any types of style augmentation or augmented domains, which is more universal and more generalized than BlindNet.
>
> 2) Completely different path to eliminate the style. BlindNet
> first augments the styles from the original image, and then
> mitigates the style variation by computing and reducing the style similarity between the per- and post- augmented styles.
> In contrast, the proposed method decouples the style and content from the frozen VFM features by high- and low- frequency components, respectively.
> Afterwards, the low-rank adaptation on the high-frequency representation is implemented with an instance normalization transformation, so that the high-frequency representation is invariant to the style shift and better generalized to unseen target domains.
>
> 3) Significant performance improvement. The official report of BlindNet on $C \rightarrow \{C, B, M, S\}$ is 38.56\%, 34.51\%, 40.11\% and 25.64\% mIoU, while this paper achieves 68.23\%, 61.94\%, 68.09\% and 50.36\% mIoU.
> The official report of BlindNet on $C \rightarrow \{B, M, G, S\}$ is 59.27\%, 71.10\%, 58.11\% and 40.43\% mIoU, while this paper achieves 65.12\%, 75.86\%, 63.78\% and 49.75\% mIoU.
>
> We will discuss this work accordingly.

---

> > ### Comment · Reviewer_4xQk · 2024-08-12
> >
> > Thanks to the efforts put in by the author for rebuttal. These answers have solved my concerns and I am raising my score to 5: Borderline accept.

---

> > > ### Author Response · Authors · 2024-08-12
> > > **Re: Official Comment by Reviewer 4xQk**
> > >
> > > We are glad to see your questions resolved, and appreciate your positive feedback after the rebttual.
> > > We will improve our work carefully per your suggestion.

---

### Official Review · Reviewer_nB2o · 2024-07-11

**Soundness:** 2
**Presentation:** 3
**Contribution:** 2
**Rating:** 5
**Confidence:** 1

**Summary:**

This paper proposes a Frequency Adaptive (FADA) learning approach. Its core idea is to process content and style information separately, by using frequency tokens. Specifically, the FADA comprises two branches for low-frequency and high-frequency components. The high-frequency components learn scene styles and eliminate their impact on DGSS, while the lower-frequency components contribute to stabilizing scene contents. Experiments in various DGSS settings demonstrate FADA's performance and versatility across different VFMs.

**Strengths:**

1. The overall presentation is pretty good.
2. The idea of using frequence token is impressive.
3. The achieved performance seems meaningful.

**Weaknesses:**

Honestly, I do not know well about this topic.
So I would like to ask some questions from the general perspective.

1. Is the assumption of "similar content yet varied style" practical? It seems that the main target of this paper is autonomous driving. Then it is rational to assume almost the same categorical distribution would be observed over various data, including the unseen test one. However, how about the joint distribution of content and style? Can we assume that they are independent across the dataset?

2. Can I see the results on the other VFMs? I understand that DINO-V2 is one of the most popular choices, but I am personally interested in whether the proposed method functions well on a more segmentation-oriented model, such as SAM. I think replacing DINO-V2 with the SAM image encoder can be an interesting experiment.

**Questions:**

See the weaknesses.

**Limitations:**

Yes.

---

> ### Author Rebuttal · Authors · 2024-08-06
>
> **Q1**: Is the assumption of "similar content yet varied style" practical? The joint distribution of content and style? Can we assume that they are independent across the dataset?
>
> **R**: Thanks for your insightful comment so that we could have a chance to further clarify the basic assumption in domain generalized semantic segmentation (DGSS).
> In the context of DGSS and autonomous driving, it has been well established and recognized by the prior DGSS works that the key challenge is 'similar content yet varied style' [13,28,33,47,66,67].
>
> Just as the reviewer acknowledged, despite the style shifts greatly in driving scenes, it is reasonable that in driving scenes the categorical distribution is nearly the same.
>
> On the other hand, we definitely agree with the reviewer that, it is a great perspective to discuss if the joint style and content distribution (i.e., the driving scenes) is independent or not.
> It should be highlighted that, the joint style and content distribution in DGSS, same as the assumption in generic visual domain generalization, is not (rigorously) independent.
> As extensively discussed in [5,67], the style in driving scenes and DGSS is impacted and jointly determined by multiple extrinsic factors such as weather, lighting, urban landscape and etc.
> An example is that, two datasets collected from two cities with different landscapes are under the same weather, which makes the style vary but not rigorously independent between each other.
>
> To sum up, given that domains in DGSS are not rigorously independent and generally have the distribution shift, which in lines with the generic visual domain generalization, we would like to raise the reviewer's attention that, the established assumption 'similar content yet varied style' [13,28,33,47,66,67] is practical in DGSS.
>
> **Q2**: Results on the other VFMs (e.g. SAM).
>
> **R**:
> We would like to kindly raise the reviewer's attention that, the results on other VFMs have been provided in Table 4 and we attach it as follows.
>
> Table R1: Performance Comparison of the proposed FADA with baseline on other VFMs. Evaluation metric mIoU in \%.
> | Backbone | Method | Citys | BDD | Map | Avg.  |
> |----------|----------|----------|----------|----------|----------|
> | SAM | Full | 57.6 | 51.7 | 61.5 | 56.9 |
> | SAM | Freeze | 57.0 | 47.1 | 58.4 | 54.2 |
> | SAM | REIN [58] | 59.6 | 52.0 | 62.1 | 57.9 |
> | SAM | **FADA** | **61.0** | **53.2** | **63.4** | **60.0** |
> |----------|----------|----------|----------|----------|----------|
> | CLIP | Full | 51.3 | 47.6 | 54.3 | 51.1 |
> | CLIP | Freeze | 53.7 | 48.7 | 55.0 | 52.4 |
> | CLIP | REIN [58] | 57.1 | 54.7 | 60.5 | 57.4 |
> | CLIP | **FADA** | **58.7** | **55.8** | **62.1** | **58.9** |
> |----------|----------|----------|----------|----------|----------|
> | EVA02 | Full | 62.1 | 56.2 | 64.6 | 60.9 |
> | EVA02 | Freeze | 56.5 | 53.6 | 58.6 | 56.2 |
> | EVA02 | REIN [58] | 65.3 | 60.5 | 64.9 | 63.6 |
> | EVA02 | **FADA** | **66.7** | **61.9** | **66.1** | **64.9** |
> |----------|----------|----------|----------|----------|----------|
> | DINOv2 | Full | 63.7 | 57.4 | 64.2 | 61.7 |
> | DINOv2 | Freeze | 63.3 | 56.1 | 63.9 | 61.1 |
> | DINOv2 | REIN [58] | 66.4 | 60.4 | 66.1 | 64.3 |
> | DINOv2 | **FADA** | **68.2** | **62.0** | **68.1** | **66.1** |
> |||||
>
> In short, when using SAM as the image encoder, the proposed method outperforms REIN [58] by 1.4\%, 1.2\% and 1.3\% mIoU on C, B and M unseen target domains;
> when using CLIP as the image encoder, the proposed method outperforms REIN [58] by 1.6\%, 1.1\% and 1.6\% mIoU on C, B and M unseen target domains;
> when using EVA02 as the image encoder, the proposed method outperforms REIN [58] by 1.4\%, 1.4\% and 1.3\% mIoU on C, B and M unseen target domains.
> These outcomes further demonstrate that the proposed method is robust and generalized on a variety of VFM image encoders.
>
> Finally, should you have further suggestions, we are glad to address during the discussion stage.

---

> > ### Comment · Reviewer_nB2o · 2024-08-10
> >
> > Thank you for the rebuttal.
> > As my concerns are clearly addressed, I would like to increase my rating for now.
> > However, as I do not know much about this topic, I think I should keep my rating in the range of borderline.
> > Probably AC will consider my opinion properly.

---

> > > ### Author Response · Authors · 2024-08-10
> > > **Re: Official Comment by Reviewer nB2o**
> > >
> > > We are glad that your concerns have been clearly addressed. We will improve our work carefully per your suggestions. Thanks again for your time and effort.

---

### Official Review · Reviewer_3Mxh · 2024-07-22

**Soundness:** 3
**Presentation:** 3
**Contribution:** 3
**Rating:** 7
**Confidence:** 3

**Summary:**

This paper introduces a novel FADA learning scheme to improve domain-generalized semantic segmentation. The proposed method leverages the Haar wavelet transformation to separate style and content information into high- and low-frequency components, respectively, allowing for better handling of domain variations. Experimental results demonstrate that FADA outperforms existing DGSS methods, showcasing its effectiveness and versatility across various vision foundation models (VFMs).

**Strengths:**

1. The proposed method is interesting and show good DGSS performances on various DG settings, which will be beneficial to the community.

2. Comprehensive ablation studies are done by the authors. The ablation on different VFMs is very important to showcase the model generalizability of the proposed low rank adaptation method.

3. The method is clearly described and great visualizations are provided.


----------------------------

Score has been updated. Thanks the authors for the rebuttal.

**Weaknesses:**

1. Lack of comparison with other recent low rank adaptation methods during the experiments, e.g., the works as follows,

[1] Gao, Z., Wang, Q., Chen, A., Liu, Z., Wu, B., Chen, L., & Li, J. (2024). Parameter-Efficient Fine-Tuning with Discrete Fourier Transform. arXiv preprint arXiv:2405.03003.
[2] Yang, Y., Chiang, H. Y., Li, G., Marculescu, D., & Marculescu, R. (2024). Efficient low-rank backpropagation for vision transformer adaptation. Advances in Neural Information Processing Systems, 36.


2. Lack of insights in the experimental analysis, e.g., in section 5.2, the authors simply listed out the performance gains brought by the proposed method compared with the state-of-art methods without mentioning why it can achieve these superior performances.

3. The ground truth images are not listed on Figure 5 and Figure 6, which hinders the comparison regarding the qualitative performances.


4.Lack of description of the implementation details. What losses are used during the experiments? Which lr scheduler is used?

**Questions:**

1. Can the proposed low rank adaptation method outperforms the other recent low rank adaptation methods as mentioned in weakness 1?


2. The authors are encouraged to add more insights towards the proposed method in the experiment analysis section.

3. The ground truth figures are suggested to be added in Figure 5 and Figure 6-

4. The implementation details are encouraged to be enriched.

**Limitations:**

yes it is mentioned be the authors.

---

> ### Author Rebuttal · Authors · 2024-08-06
>
> **Q1**: Compare recent LoRA methods [a,b].
>
> [a] Gao, Z., et al. (2024). Parameter-Efficient Fine-Tuning with Discrete Fourier Transform. ICML.
>
> [b] Yang, Y., et al. (2024). Efficient low-rank backpropagation for vision transformer adaptation. NeurIPS.
>
> **R**: We've compared both methods, namely, FourierFT [a] and WHT [b], with the baseline, REIN [58] and the proposed FADA under the G$\rightarrow$ C, B, M setting.
> For fair evaluation, both [a] and [b] are attached to process the VFM features of each layer as the common low-rank adaptation paradigm does.
> The results in Table R1 show that the proposed FADA shows its superiority than the recent methods [a, b] on all unseen target domains with a $\textgreater$ 2\% mIoU improvement.
>
> Table R1: Performance Comparison of the proposed FADA with recent PEFT methods [a,b]. Evaluation metric mIoU in \%.
> | Method | Citys | BDD | Map | Avg.  |
> |----------|----------|----------|----------|----------|
> | Full | 63.7 | 57.4 | 64.2 | 61.7 |
> | Freeze | 63.3 | 56.1 | 63.9 | 61.1 |
> | LoRA [24] | 65.2 | 58.3 | 64.6 | 62.7 |
> | REIN [58] | 66.4 | 60.4 | 66.1 | 64.3 |
> | FourierFT [a] | 66.1 | 59.2 | 65.8 | 63.7 |
> | WHT [b] | 65.8 | 58.9 | 65.3 | 63.3 |
> | FADA | **68.2** | **62.0** | **68.1** | **66.1** |
> |||||
>
> **Q2**: Sec.5.2, the authors are encouraged to add more insights towards the proposed method in the experiment analysis section.
>
> **R**:
> Thanks for your constructive suggestions, so that we could have the chance to provide more insight in the experimental section. We will enrich the following analysis accordingly.
>
> (1) From the perspective of representation. Vision Transformer based encoder is more capable to capture the scene content owing to the long-range dependencies than the CNN based encoder, which explains the significant improvement of Transformer based methods than CNN based methods.
> On top of it, both the proposed method and REIN [58] use Transformer based foundation model, which not only possesses the inherited representation ability from the Transformer, but also occupies the inherited out-of-distribution generalization ability from the large-scale data pre-training.
> Most importantly, compared with REIN [58], the proposed method further advances low-rank adaptation in the frequency space, which handles the scene content and the style variation in a divide-and-conquer manner, therefore further improving the generalization ability on unseen target domains.
>
> (2) From the perspective of domain gap. The first two experiments use GTA5 and SYNTHIA as source domains, respectively.
> Both source domains are synthetic data.
> Even if trained on such synthetic source domain, the proposed method still shows a generalization performance of 68.2\% on real-world CityScapes target domain.
> This outcome is very close to some modern CNN and Transformer segmentation models when full-trained on CityScapes, which further indicates the generalization ability of proposed method.
> In contrast, the compared DGSS methods still shows a significantly inferior performance when the domain gap is huge.
>
> (3) From the perspective of low-rank adaptation in frequency space. In visual domain generalization and DGSS, an ideal representation is to be stable to the scene content when the cross-domain styles shift greatly.
> As many prior works [25,36,32,62,56] reflect, the decoupling of high- from low-frequency component is an important research line to decouple the style from content.
> The proposed method takes advantage of this aspect, and implements Haar wavelet transformation to realize this objective.
> The individual low-rank adaptation for low- and high- frequency components provides a divide-and-conquer path for VFM features to handle the style and content, respectively, and therefore significantly improves the performance on unseen target domains when compared with REIN [58].
>
> Finally, we would like to kindly raise the reviewer's attention that, in Sec.5.3 we have provided the ablation studies from four different aspects to analyze why the proposed method achieves a better performance than the VFM baseline and REIN [58].
>
> **Q3**: The ground truth images are not listed on Figure 5 and Figure 6, which hinders the comparison regarding the qualitative performances.
>
> **R**: Thanks for your constructive suggestions.
> We have accordingly provided the ground truth map for each sample in Fig.5 and Fig.6.
> The updated Fig.5 and Fig.6 have been provided in the attached PDF file in the general response, named as Fig.R2 and Fig.R3, respectively.
>
> **Q4**: Lack of description of the implementation details. What losses are used during the experiments? Which lr scheduler is used?
>
> **R**:
> We are sorry that the implementation details of the proposed method is extensive than intended.
> In general, all the loss functions, learning scheduler and other hyper-parameters keep the same as REIN for fair comparison.
> Specifically, the optimizer uses AdamW.
> The initial learning rate for the backbone is $1\times10^{-5}$.
> The initial learning rate on the decoder and the proposed FADA's parameter is set $1\times10^{-4}$.
> The final loss function $\mathcal{L}$ directly inherit the losses from the Mask2Former decoder [12], given by
>
> $\mathcal{L} = \lambda _ {ce} \mathcal{L} _ {ce} + \lambda _ {dice} \mathcal{L} _ {dice} + \lambda _ {cls} \mathcal{L} _ {cls}$
>
> where $\mathcal{L} _ {ce}$, $\mathcal{L} _ {dice}$ and $\mathcal{L} _ {cls}$ denote the cross-entropy loss, dice loss and classification loss.
> Here the hyper-parameters $\lambda _ {ce}$, $\lambda _ {dice}$ and $\mathcal{L} _ {cls}$ are 5.0, 5.0 and 2.0, respectively.
> For more details of the classification loss, please refer to [12].
> We will enrich these details accordingly.
>
> Finally, should you have further questions and suggestions, we are glad to address during the discussion stage.

---

> > ### Comment · Reviewer_3Mxh · 2024-08-08
> > **Response to the author**
> >
> > Thank you for your effort during the rebuttal. My concerns are addressed by the authors, thereby I would like to improve my rating to 7.

---

> > > ### Author Response · Authors · 2024-08-08
> > > **Re: Response to Reviewer 3Mxh**
> > >
> > > We would like to express our gratitude to be recognized by the reviewer, and are glad to see your questions resolved. We will improve our work carefully per your suggestions.

---

### Official Review · Reviewer_eWsS · 2024-07-29

**Soundness:** 3
**Presentation:** 3
**Contribution:** 3
**Rating:** 6
**Confidence:** 4

**Summary:**

This paper focuses on adapting the vision foundation model for domain-generalized segmentation via frequency-aware adaptation. Specifically, the intermediate features are decomposed into low-frequency (content) and high-frequency (style) components, which are then being processed with separate low- and high- frequency branches. To this end, sufficient experiments on extensive datasets validate the effectiveness of the proposed method.

**Strengths:**

The problem setting is important. To adapt the vision foundation model in a domain-generalizable manner, parameter-efficient fine-tuning such as LoRA has been extensively used, this paper proposed a solution in an orthogonal direction -- adapting in the frequency domain. The core intuition makes sense to me. This paper decomposes the features into low-frequency and high-frequency domains and is adapted separately. The whole solution is simple and elegant. Sufficient comparison on extensive datasets and difference foundation model architectures validates the effectiveness of the proposed method.

**Weaknesses:**

1. When conducting the adaptation, it is intuitive that the low-frequency/ high-frequency components correspond to the content/style. However, can the author comment on the intuition on adapting in all the transformer laters? It may not be that clear whether the low-frequency and high-frequency features in the later layers have such correspondence anymore. From that sense, it would be nice to ablate the layer location where the frequency adapter is added (as opposed to being added to all the layers)


2. When comparing with the REIN, REIN only has a trainable parameter of 2.99M (compared to FADA 11.65), the author also shows in Table 3 that FADA performance is influenced by the number of Rank and tunable parameters. It would be nice to add an experiment that compares these two methods with similar tunable parameters.


3. The core intuition behind the decomposition is to prioritize the domain-invariant low-frequency features (content features) over the high-frequency features. However, as indicated by the ablation results in Table 2, better outcomes were achieved by adapting to all features. The current model design does not differentiate between high-frequency and low-frequency features, raising questions about the alignment between the model design and the initial intuition. A potential experimental setup that might clarify this involves training the model on multiple domain datasets with a shared low-frequency branch and distinct high-frequency branches. However, this would alter the experimental conditions. I would appreciate the authors' comments on this matter.

4. Conducing Haar transform can cause additional inference time, it would be nice to include the inference time comparision.

**Questions:**

See the weakness section.

**Limitations:**

Limitations have been discussed.

---

> ### Author Rebuttal · Authors · 2024-08-06
>
> **Q1**: 1) Intuition on adapting in all the transformer layers. 2) Ablate the layer location.
>
> **R**: Thanks for the constructive comment.
>
> 1) The high-level idea of our intuition focuses on low-rank adaptation (LoRA) in the frequency space.
> As existing LoRA methods [24,58] implements the adaptation on each of the transformer layer, and we further introduce the frequency space.
> Therefore, we treat the frequency space as a whole, and embed it into each of the transformer layer.
>
> 2) But we definitely agree with the reviewer that, it would be meaningful to have an in-depth analysis on the learning behaviour from shallow to deep.
> To this end, apart from the baseline (REIN [58]) and the proposed method, we further provide two experiments, where the frequency adapter is attached in the first half seven layers (denoted as shallow) and the second half seven layers (denoted as deep), respectively.
> Results in Table R1 show that:
> - using the frequency adapter on the first half layers (shallow) shows a slightly better performance than on the second half layers (deep). It may be explained that the shallower features contain more cross-domain styles, such as illumination, landscape and etc.
> - using the frequency adapter on all layers (ours) achieves the best performance, indicating its effectiveness on all layers.
>
> Table R1: Ablation study on the positions of the frequency adapters. GTA5 as the source domain. CityScapes, BDD and Mappilary are unseen target domains. Evaluation metric mIoU in \%.
> | Method | Citys | BDD | Map | Avg.  |
> |----------|----------|----------|----------|----------|
> | Full | 63.7 | 57.4 | 64.2 | 61.7 |
> | Freeze | 63.3 | 56.1 | 63.9 | 61.1 |
> | REIN [58] | 66.4 | 60.4 | 66.1 | 64.3 |
> | Shallow | 67.6 | 61.5 | 67.4 | 65.5 |
> | Deep | 67.3 | 61.2 | 67.0 | 65.2 |
> | FADA | **68.2** | **62.0** | **68.1** | **66.1** |
> |||||
>
> **Q2**: Compare FADA with REIN under similar tunable parameters.
>
> **R**: Thanks for your constructive suggestion.
> The feature dimension of MLPs in LoRA impacts the parameter number. We therefore minimize the feature dimension in FADA, from 1024 to 256, so that the parameters are reduced significantly.
> Results in Table R2 show that, even if the feature dimension is only one fourth of REIN [58], FADA still shows a 1.1\% mIoU improvement on three unseen domains.
>
> Table R2: Comparison between FADA and REIN under similar tunable parameters. GTA5 as source domain.
> | Method | Rank | Dimension | #para. | Citys | BDD | Map | Avg.  |
> |----------|----------|----------|----------|----------|----------|----------|----------|
> | REIN | 16 | 1024 | 2.99M | 66.4 | 60.4 | 66.1 | 64.3 |
> | **FADA** | 16 | 256 | **2.92M** | **67.6** | **61.2** | **67.5** | **65.4** |
> ||||||||
>
> **Q3**: 1) prioritize low- over high-frequency features v.s. Table 2 adapting to all features. 2) suggest an experiment design.
>
> **R**: Thanks for your constructive comments, so that we could have a chance to have a more in-depth analysis between the low- and high- frequency representation.
>
> 1) In general, the intuition of the proposed method is to allow the high-frequency components to be invariant to the style shift, instead of eliminating the entire high-frequency components.
> The reasonable behind is that, high-frequency components of an image representation consist of not only styles, but also other types of information such as the scene structure, object boundaries and etc.
> Scene structure and object boundary is also important for scene segmentation.
> Therefore, our objective is to adapt the high-frequency components so that they are invariant to the cross-domain shift, not to directly eliminate all the high-frequency components.
>
> 2) We definitely agree with the reviewer that, it would be beneficial to specify the high-frequency features from multiple domains.
> Following your suggestion, we follow the evaluation setting where GTA5 and Synthia are both used as the source domains, while CityScapes, BDD and Mapillary are used as unseen target domains, and REIN [58] as the baseline.
> After Haar wavelet transformation on the per-batch image from each domain, their low-frequency components are fused into a single low-rank adaptation module.
> Instead, two low-rank adaptation modules are used to process the high-frequency components from the two source domains, respectively (denoted as $HH_{G+S}$).
> Both high-frequency components are implemented with an instance normalization after the adaptation.
> It is compared with the scenario where one high-frequency component is used, denoted as $HH_G$ and $HH_S$.
> The results in Table R3 show that, handling the high-frequency component from each source domain indeed improves the performance on unseen target domains.
> Handling the high-frequency components from both source domains shows a further improvement.
>
> Table R3: Impact on the learning paradigm of high-frequency components.
> GTA5 and SYNTHIA are both source domains. CityScapes, BDD and Mapilary are unseen target domains.
> | Method | Citys | BDD | Map | Avg.  |
> |----------|----------|----------|----------|----------|
> | Freeze | 64.8 | 60.2 | 65.2 | 63.4 |
> | REIN [58] | 68.1 | 60.5 | 67.1 | 65.2 |
> | $HH_{G}$ | 68.9 | 61.4 | 68.0 | 66.1 |
> | $HH_{S}$ | 68.7 | 61.2 | 67.6 | 65.8 |
> | $HH_{G+S}$ | 69.4 | 61.8 | 68.4 | 66.5 |
> | **FADA  (ours)** | **70.2** | **62.4** | **68.9** | **67.2** |
> |||||
>
> **Q4**: Compare the inference time.
>
> **R**: We compare the inference time of the proposed FADA along with REIN [58] and a vanilla VFM. All the experiments are conducted on a single RTX 3090Ti GPU.
> As reported in Table R4, the proposed FADA, implementing Haar wavelet transformation with additional parameters, only leads to a slight decrease on the inference time than the previous REIN [58].
>
> Table R4: Inference Time Comparison. Evaluation Metric: Frame Per Second (FPS).
> | Method | Vanilla VFM | REIN | FADA (Ours)  |
> |----------|----------|----------|----------|
> | FPS | 2.50 | 2.49 | 2.46 |
> ||||

---

> > ### Comment · Reviewer_eWsS · 2024-08-12
> >
> > Thank the author for the detailed feedback, this solves most of my concerns. I will keep the original rating to reflect my confidence of this paper

---

> > > ### Author Response · Authors · 2024-08-12
> > > **Re: Official Comment by Reviewer eWsS**
> > >
> > > We would like to express our gratitude to be recognized by the reviewer, and are glad that your concerns have been resolved.
> > >
> > > We will improve our work carefully per your suggestions.

---

### Official Review · Reviewer_esrP · 2024-07-30

**Soundness:** 3
**Presentation:** 3
**Contribution:** 2
**Rating:** 4
**Confidence:** 4

**Summary:**

This paper leverages the vision foundation model (VFM) for domain-generalized semantic segmentation (DGSS). It proposes to adapt the VFM to the downstream task in the frequency domain. The proposed method decouples the low-frequency and high-frequency components of the VFM features and separately applies low-rank adaptation to them. This aims to stabilize the domain-invariant content and, on the other hand, eliminate the impact of domain-specific styles on DGSS.

**Strengths:**

* The paper is well-written and organized.
* The idea of applying low-rank adaptation of VFM in the frequency domain has some certain novelty.
* The proposed method achieves competitive performance on several datasets across different settings compared to previous methods.

**Weaknesses:**

* **Insufficient justification of key concepts:** The paper's claim that domain-specific style and domain-invariant content can be demonstrated by high-frequency and low-frequency components is not well-justified in this context. And, the explanation of how the proposed method stabilizes domain-invariant content while mitigating domain-specific style is insufficiently supported.

* **Lack of detailed analysis on style elimination:** Given the stated importance of style-invariant properties in VFM for DGSS representations, eliminating the impact of domain-specific styles is crucial. While the paper uses instance normalization to address this, it lacks a detailed analysis of the effectiveness of this approach.

* **Unclear representation of Haar components:** Although the paper suggests that the three high-frequency components represent domain-specific style information that should be eliminated for better DGSS, Table 2 indicates that the best results are achieved by preserving all components. The discussion around this in Line 250 is not sufficiently informative.

* **Inadequate comparison with previous methods:** The paper also mentions previous works that have utilized FFT for frequency analysis and claims that it may be the first to use Haar wavelet for domain generalization. However, it does not adequately discuss the advantages of Haar wavelet over other methods or provide comparative analysis in this context.

**Questions:**

VFMs are known for their strong generalization ability across different domains. What specific style-invariant properties of VFM are the authors attempting to exploit in this paper?

**Limitations:**

Please provide the necessary evidence and adequate clarification of the problems in the weaknesses.

---

> ### Author Rebuttal · Authors · 2024-08-06
>
> **Q1**: Justify: 1) domain-specific style and domain-invariant content are demonstrated by high- and low-frequency components 2) how the proposed method stabilizes domain-invariant content while mitigating domain-specific style.
>
> **R**: Thanks for your valuable comment. To clarify:
>
> 1) In visual domain generalization, an important research line is to learn a domain-invariant content representation from the frequency feature space. It has been observed and well-documented that the low- and high- frequency components rest more domain-invariant content and domain-specific style properties [25,36,32,62,56].
>
> 2) To validate this point in our context, we extract the frozen VFM feature from driving scene samples containing the same semantic categories (i.e., content) but from four unseen domains (i.e., style), and implement the Haar wavelet transformation on them.
> According to the t-SNE feature visualization in Fig.R1 Row 1 (attached in the 1-pg PDF file), the high-frequency features ($HH$, $HL$, $LH$) exhibit some domain-specific clusters, where driving scenes from the same domain (points of the same color) lean to cluster together.
> In contrast, the low-frequency features ($LL$) allow the driving scenes from different domains (point of different color) to be more uniformly mixed and distributed.
>
> 3) To inspect the effectiveness of the proposed method on mitigating the domain-specific features, we implement the instance normalization transformation on the high-frequency features ($HH$, $HL$, $LH$) from the VFM.
> After that, the t-SNE feature visualization is conducted and the results are shown in Fig.R1 Row 2.
> It can be seen that, compared with the original high-frequency features in Fig.R1 Row 1, the domain-specific clusters are alleviated, and the samples from different domains (points of different colors) are more uniformly distributed.
>
> **Q2**: More analysis on instance normalization for style invariance.
>
> **R**: Thanks for your insightful comments. Please refer to the 2) and 3) response to Comment\#1 for the detailed analysis.
>
> In addition, we would like to kindly raise the reviewer's attention that, Fig.3 and Fig.4 in the main text provide analysis on how the proposed method eliminates the domain-specific styles in both feature space and channel-wise activation patterns.
>
> **Q3**: L250 \& Table 2. Eliminate high-frequency components or preserving all components.
>
> **R**: To clarify, our objective is not *to eliminate the high-frequency components that rest in domain-specific styles*, but *to be invariant to cross-domain style variation* (L172) \& *to decouple the high-frequency component from the impact of cross-domain styles* (L250).
> The reason behind is that the high-frequency components contain not only styles, but also other information such as structure and object boundary.
> A naive removal of all high-frequency components loses such information, which degrades a scene representation and declines the segmentation performance.
> Therefore, our objective is to allow the high-frequency representation to be invariant to the cross-domain styles by instance normalization, not to entirely eliminate all the high-frequency representation.
> It aligns with the ablation study in Table 2, where the low-rank adaptation with instance normalization on high-frequency components contribute positively to the performance.
> We will clarify this point accordingly when revising.
>
> **Q4**: Discuss/compare the advantages of Haar wavelet over FFT methods.
>
> **R**:
> 1) Both FFT and Haar wavelet decouple a scene representation into low- and high- frequency components.
> However, compared with FFT, the orthonormal basis of Haar wavelet warrants the orthonormality between the low- and high- frequency components.
> Orthonormality is essentially a type of de-correlation, which provides a better separation between the low- and high- frequency components.
> In the context of DGSS, the orthonormality of Haar wavelet transformation allows a better separation between the scene content and the style.
>
> 2) From the experimental side,  please refer to Table R1 in the general response for the comparison.
> It is observed that, the proposed method, with the aid of Haar wavelet, shows a predominant performance improvement than FFT based [a] on all unseen target domains.
> Meanwhile, we respectively ask for the reviewer's understanding that, as most frequency decoupling based domain generalization methods are devised for classification task , it would be difficult and impractical to adapt all these methods for VFM based DGSS.
>
> [a] Gao, Z., Wang, Q., Chen, A., Liu, Z., Wu, B., Chen, L., \& Li, J. (2024). Parameter-Efficient Fine-Tuning with Discrete Fourier Transform. ICML 2024.
>
> **Q5**: What specific style-invariant properties?
>
> **R**: Indeed, VFM has an inherited ability to generalize to out-of-distribution.
> However, the performance of current large segmentation models still lacks in specific down-stream scenarios, so additional training is needed.
> Consequently, fine-tuning VFM on DGSS encounters the unique challenge of DGSS, that is, an ideal representation for the task of DGSS is to learn stable pixel-wise content representation despite the cross-domain style shift [44,26,13,45,63,60,47].
> Here the style shift can be posed by different types of weather, illumination, urban landscape and etc [67].
> However, the fine-tuning of VFM is highly dependent on the low-dimensional intrinsic embeddings [58], which is subtle and fragile to the distribution shift, which is posed by the style shift in the context of DGSS.
> Therefore, an ideal VFM representation for DGSS is supposed to demonstrate robust style-invariant property to generalize well on different types of weather, illumination, urban landscape and etc [67].
>
> On the other hand, as shown in Table 1, 2, 4 and 5, the modified VFM representation from the proposed method shows a significantly better generalization on unseen target domains, in different weather and day-night conditions.

---

### Author Rebuttal · Authors · 2024-08-06

We thank the reviewers for their time and constructive suggestions, and are glad that the reviewers unanimously give appreciation in a few points:

- Technique Contribution \& Innovation (**esrP**: low-rank adaptation in the frequency domain has novelty; **eWsS**: the whole solution is simple and elegant; **3Mxh**: the proposed method is interesting; **nB2o**: the idea of using frequency token is impressive; **ojsk**: compelling.)

- Writing \& Presentation (**esrP**: well-written and organized; **eWsS**: good presentation; **3Mxh**: The method is clearly described
and great visualizations are provided; **nB2o**: The overall presentation is pretty good; **ojsk**: well-written.)

- Good Performance (**esrP**: competitive performance; **eWsS**: Sufficient comparison; **3Mxh**: how good DGSS performances; **nB2o**: The achieved performance seems meaningful; **4xQk**: good performance; **ojsk**: consistent improvement.)

However, there are also some major concerns as follows.

- Compare with frequency based / parameter-efficient fine-tuning methods (**esrP** comment\#4; **3Mxh** comment\#1; **4xQk** comment\#2).

**R**: A recent frequency based foundation model fine-tuning method FourierFT [a] and a low-rank adaptation method WHT [b] are involved for comparison.
While some improvement over baseline can be observed, the proposed FADA still demonstrates the optimal performance in DGSS.
Meanwhile, we respectively ask for reviewer's understanding that, most existing frequency decoupling methods are devised for cross-domain classification [25,36,32,62,56], and can be difficult to be incorporated into the foundation model fine-tuning pipelines.

Table R1: Performance Comparison of the proposed FADA with recent PEFT methods [a,b]. Evaluation metric mIoU in \%.
| Method | Citys | BDD | Map | Avg.  |
|----------|----------|----------|----------|----------|
| Full | 63.7 | 57.4 | 64.2 | 61.7 |
| Freeze | 63.3 | 56.1 | 63.9 | 61.1 |
| LoRA [24] | 65.2 | 58.3 | 64.6 | 62.7 |
| REIN [58] | 66.4 | 60.4 | 66.1 | 64.3 |
| FourierFT [a] | 66.1 | 59.2 | 65.8 | 63.7 |
| WHT [b] | 65.8 | 58.9 | 65.3 | 63.3 |
| FADA | **68.2** | **62.0** | **68.1** | **66.1** |
|||||

- Analyze how handling low- and high- frequency component help stabilize content and invariant to style (**esrP** comment\#1, \#2; **ojsk** comment\#1).

**R**:
We extract the frozen VFM feature from driving scene samples containing the same semantic categories (i.e., content) but from four unseen domains (i.e., style), and implement the Haar wavelet transformation on them.
According to the t-SNE feature visualization in Fig.R1 (attached in the 1-pg PDF file), the high-frequency features ($HH$, $HL$, $LH$) exhibit some domain-specific clusters, where driving scenes from the same domain (points of the same color) lean to cluster together.
In contrast, the low-frequency features ($LL$) allow the driving scenes from different domains (point of different color) to be more uniformly mixed and distributed.

To inspect the effectiveness of the proposed method on mitigating the domain-specific features, we implement the instance normalization transformation on the high-frequency features ($HH$, $HL$, $LH$) from the VFM.
After that, the t-SNE feature visualization is conducted and the results are shown in Fig.R1 Row 2.
It can be seen that, compared with the original high-frequency features in Fig.R1 Row 1, the domain-specific clusters are alleviated, and the samples from different domains (points of different colors) are more uniformly distributed.

- Eliminating styles \& high-frequency features v.s. preserving all features (**b4sx** Comment\#5; **hFq3**: comment\#3).

**R**: We humbly suggest there may existing some misunderstanding in the methodology objective.
Eliminating *the high-frequency components* and *be invariant to the domain-specific styles* is two different things.
The high-frequency components of a scene not only contain styles, but also contain other types of information such as structure, object boundary and etc. A naive removal of all the
high-frequency components also leads to the loss of such information, which can degrade a scene representation and therefore declines the segmentation performance.
Therefore, our objective is to allow
the high-frequency representation to be invariant to the cross-domain styles by instance normalization,
not to entirely eliminate all the high-frequency representation. It aligns with the ablation study in
Table 2, where the low-rank adaptation with instance normalization on high-frequency components
can contribute positively to the performance. We will clarify this point accordingly when revising

We hope our clarification could help to make a more informed evaluation to our work.
In the following individual response, we provide answers to each raised weakness/question.

Best regards,

Authors

---

### Decision · Program_Chairs · 2024-09-25

**Decision:**

Accept (poster)

**Comment:**

Dear authors,
Overall this draft got positive reviews (R:5 / C:5, R:4 / C:4,  R:6 / C:4, R: 7 / C:3,  R:5 / C:2,  R:5 / C:1 ). Two reviewers updated their rating to positive side.
After careful review of the reviewer comments, answers by the authors, and draft itself, we are accepting the draft. However, authors are strongly encouraged to improve the readability of the paper, incorporate suggestions by the reviewers and handle the comments.
For example,
1) "advantages of Haar wavelet over other methods or provide comparative analysis in this context."  --- Needs to be  added with clarity and given due importance in the. text.
2) Role of Instance Normalization for style invariance needs more explanation and study (even if added to supplementary material and referred in the text).
3) Fig.1's. caption should include basic information to read the figure. LL, HH, etc.. are defined much below in the text rather than before. Therefore, add this information in the caption.  At the same time  statements like "Domain Generalized Semantic Segmentation (DGSS) learns to generalize to arbitrary unseen target domains when only trained on the source domain; The key challenge lies in the stability
of the scene content, while the domain gap is caused by the style ..."  should be part of the text rather than caption.
4) Table-1 either explain what is C, B and M are or explicitly refer to the text where these are properly defined.
5) incorporate experiments using to reply to the author's questions.
etc..